# Exploring the cecal microbial community associated with fat deposition in sheep and its possible pathways of action

Jiangbo Cheng,[1] Dan Xu,[1] Deyin Zhang,[1] Kai Huang,[1] Yukun Zhang,[1] Xiaolong Li,[1] Yuan Zhao,[1] Liming Zhao,[1] Quanzhong Xu,[1] Xiaobin Yang,[1] Zongwu Ma,[1] Huibin Tian,[1] Xiaoxue Zhang,[2] Weimin Wang[1]

**ABSTRACT** Fat deposition is a crucial economic trait during sheep growth and development, closely linked to economic returns. Current research on sheep's digestive tract predominantly focuses on the rumen, but the composition of the cecal microbiota and its relationship with host fat deposition remains largely unexplored. In this study, we sequenced the cecal microbiota of 60 Hu sheep, exhibiting marked differences in traits. The most abundant species in the sheep cecum were Firmicutes and Bacteroidota. Statistical analyses revealed significant differences in microbial community structures among different fat-deposition groups ($P < 0.05$). Using a random forest regression model and linear regression, 15 microbial biomarkers, including *Lachnospiraceae_NK3A20_group*, *Turicibacter*, and *Bacteroides*, were identified as key contributors to fat deposition. Additionally, volatile fatty acids (VFAs) in the cecum and biochemical indices in serum were measured. Acetic acid was the most abundant VFA in the cecum, while isobutyric acid levels were significantly higher in the low-fat group than in other groups ($P < 0.05$). Serum triglyceride (TG) levels were significantly higher in the high-fat group ($P < 0.05$). Correlation analysis revealed a significant association between *Lachnospiraceae_NK3A20_group* and acetic acid levels, as well as between TG levels and fat deposition traits ($P < 0.05$). TG levels were negatively correlated with acetic acid concentrations ($P < 0.05$). These findings suggest that the cecal microbiota influences fat deposition in sheep, potentially via the VFAs-TG metabolic pathway.

**IMPORTANCE** Compared with muscle development, fat deposition consumes more energy, and controlling the fat deposition process can effectively reduce energy waste. Current research on rumination mainly focuses on the rumen but lacks research on the hindgut. This study identifies differences in the cecal microbiota of sheep with varying fat deposition levels and highlights significant correlations between specific microorganisms, cecal metabolites, and host traits. Therefore, the regulation of cecal microorganisms can help improve the fat deposition characteristics of sheep.

**KEYWORDS** Hu sheep, fat deposition, cecal microorganisms, triglycerides, volatile fatty acids

Sheep (*Ovis aries*), one of the first livestock species to be domesticated, are now globally widespread and serve as a significant source of meat-derived protein. Among them, the fat-tailed sheep breed, constituting approximately 25% of the global sheep population, evolves from thin-tailed wild sheep through natural and artificial selection (1). This breed's superior fat deposition capability enhances its resistance to environmental stress. During the grazing phase, fat-tailed sheep accumulate more fat in seasons of plentiful pasture to sustain energy through harsh conditions (2). However, with the increase in market demand, sheep breeding gradually transforms into intensive industrialization. Furthermore, reducing fat accumulation effectively accelerates muscle

**Peer Reviewer** Jian Zhou, Chinese Academy of Sciences, Beijing, China

Address correspondence to Weimin Wang, wangweimin@lzu.edu.cn.

The authors declare no conflict of interest.

See the funding table on p. 14.

growth, reduces feed costs, and increases production income (3). At the same time, studies show that meat products with excessive fat content in the human body cause obesity, diabetes, and other related metabolic diseases, which seriously endanger human health. Therefore, the demand for lean products in the consumer market increases (4, 5). With the market increasingly favoring quality over quantity, it is crucial to explore methods to control fat deposition in sheep.

The regulation of fat deposition involves various factors. Currently, many studies analyze this process from the perspective of genomic inheritance (6, 7). However, our preliminary research results show that the growth traits are regulated by two channels: genetic variation and digestive tract microflora (8).

In the digestive tract of monogastric animals, the cecum is the most critical fermentation site due to high bacterial diversity, making it a primary focus of research (9, 10). For ruminants, the rumen serves as the main fermentation site, and volatile fatty acids (VFAs) are obtained and supplied to the host through the metabolism of microorganisms in the rumen. However, the digestive tract of ruminants is a complex system, and the composition of microbial flora is spatially heterogeneous (11). The microbial flora in the rumen is well characterized (12), but the hindgut is less studied. Studies show that the rumen has the highest concentration of VFAs in the digestive tract of ruminants, followed by the cecum (13). The most basic function of VFAs in ruminants is to provide energy, and excess energy is converted into fat and stored. At the same time, VFAs regulate fat production in tissues by regulating the secretion of related hormones and gene expression (14). Gastrointestinal microorganisms play a regulatory role in gastrointestinal health and peripheral tissues such as adipose tissue through their metabolite VFAs (15). Currently, grain-pellet feed is used in large-scale sheep breeding because it improves production efficiency (16). Compared with the foregut, grain feeding has a greater impact on hindgut microorganisms, and the imbalance of hindgut flora has an impact on the economic traits of the host (17). Therefore, understanding the cecal microbiota and its role in fat deposition is essential.

We hypothesize that there are key differential bacterial flora in the cecum of sheep with different levels of fat deposition, and there is an interactive relationship between the differential bacterial flora and the host, which can regulate the host's fat deposition traits. In this study, the fat deposition-related traits of 303 Hu sheep were measured and grouped based on fat deposition traits. Using 16S rDNA sequencing, we investigated the cecal microbiota associated with different levels of fat deposition and their potential functions, providing insights for regulating this process in sheep.

## MATERIALS AND METHODS

### Animals and sampling

In this study, 303 male Hu sheep with similar birth dates were raised at Minqin Defu Agricultural Technology Co., Ltd. (Gansu, China). All sheep were vaccinated according to standard protocols, weaned at 56 days, and subsequently housed in separate enclosures. The sheep had unrestricted access to water and were fed a diet of full-priced pellet feed, detailed in Table S1. Following weaning, the lambs underwent a 24-day adaptation period during which the proportion of alfalfa silage in their diet was gradually reduced until completely replaced by pellet feed. Performance assessments were conducted at 80 days, with weight gain and feed intake monitored from 80 to 180 days. Growth traits, including body length, height, and chest circumference, were measured at 180 days. Body mass index (BMI), average daily feed intake (ADFI), average daily gain (ADG), and feed conversion rate were calculated. Based on previous reports, the formula for sheep BMI is weight (kg) divided by the square of the body length (m) (18). The feeding regimen and environmental conditions were consistent throughout the study. From the initial 303 sheep, 60 (20%) were selected as the test population size for subsequent experiments. Based on BMI rankings, 20 sheep from the high and low extremes were selected, while 20 sheep from the median served as the control group. By specifically

selecting individuals at the high and low extremes of BMI (along with a middle BMI reference group), we aimed to enhance the ability to discern distinct differences in microbiota composition and metabolic traits. The inclusion of the middle BMI group provided a baseline for comparison between the extremes.

When the experimental group reached 180 days, 5 mL of venous blood was collected, and serum was separated and stored in a −80°C ultra-low temperature refrigerator for subsequent determination of biochemical indicators. After euthanasia, the weights of tail fat, perirenal fat, and mesenteric fat were recorded. A portion of the visceral fat was preserved in 4% paraformaldehyde for histological examination. Immediately collect the entire cecum from the experimental animals, ligate both ends to prevent the contamination of the intestinal contents, and collect samples of the intestinal contents from the same region. The samples should be rapidly frozen in liquid nitrogen and stored in a −80°C ultra-low temperature freezer after sampling is completed.

## Sample determination

The contents of fat, moisture, salt, and protein in the meat samples were quantified using a FoodScan 2 meat quality analyzer (Fosihua Science and Trade Co., Ltd, Beijing, China). VFAs in cecal chyme were determined by gas chromatography (Agilent Technologies, Inc., Santa Clara, CA), and the specific method was described by Zhang et al. (19). Biochemical indicators in serum were measured using a fully automatic biochemical analyzer (Erba, Mannheim, Germany). The measured indexes included albumin (ALB), alkaline phosphatase (ALP), alanine aminotransferase (ALT), aspartate aminotransferase (AST), creatine kinase (CK), creatinine (CR), direct bilirubin (DBIL), glucose (GLU), lactate dehydrogenase (LDH), total bilirubin (TBIL), triglyceride (TG), and total protein (TP). Frozen sections of visceral fat were stained using Oil Red O stain, and the area of lipid droplets was measured using Image Pro Plus (version 6.0).

## DNA extraction and amplification

The total genomic DNA of the cecal samples was extracted using the cetyltrimethylammonium bromide (CTAB)/SDS method. The concentration and purity of the DNA were assessed using 1% agarose gel electrophoresis, and the DNA was subsequently diluted to a concentration of 1 µg/µL. Specific primers were used to amplify the 16S V3–V4 region (314F: CCTAYGGGRBGCASCAG and 806R: GGACTACNNGGGTATCTAAT). The reaction system for amplification was 15 µL Husion Master Mix (New England Biolabs, Massachusetts, USA), 0.2 µL forward and reverse primers, and 10 ng template DNA. PCR cycle conditions were denaturation at 98°C for 10 s, annealing at 50°C for 30 s, and elongation at 72°C for 30 s. The same volume of IX loading buffer (containing SYB green) was mixed with the PCR product, and 2% agarose gel electrophoresis was used for detection. The mixed PCR products were purified using a gel extraction kit (Qiagen, Hilden, Germany).

## Library construction and sequencing

Libraries were constructed using the TruSeq DNA PCR-Free Sample Preparation Kit (Illumina, California, USA) (20), and index codes were added. Library quality inspection was performed by Qubit@2.0 Fluorometer (Thermo Scientific, Massachusetts, USA) and Agilent Bioanalyzer 2100 system. After passing the quality inspection, the library was sequenced using the Illumina NovaSeq platform to generate 250 bp paired-end reads.

## Statistical analysis of data

After obtaining the original sequencing data, it was assigned according to the unique barcode of the sample, and the barcode and primer sequences were excised. The paired-end sequences were merged using FLASH (version 2.7) (21), and FASTP (version 0.23.4) was used to quality control the Raw Tags to obtain Clean Tags. The UCHIME algorithm was used to detect chimeric sequences, and after removing the chimeric

sequences, Effective Tags were obtained for subsequent analysis (22). QIIME2 (version 2023.5.1) was used for subsequent analysis (23). The effective tags were imported using the tools import function in QIIME2, and the denoising of valid reads was performed using the DADA2 plugin to generate amplicon sequence variants (ASVs) and their representative sequences with the supplementary parameters "denoise-single --p-trunc-len 400 --p-trim-left 20." A phylogenetic tree was constructed using the phylogeny align-to-tree-mafft-fasttree function, and sample microbial diversity indices were calculated using the diversity core-metrics-phylogenetic function. Species annotation (phylum, class, order, family, and genus) was performed based on the SILVA 138 database (https://www.arb-silva.de/documentation/release-138/) using the feature-classifier classify-consensus-blast function with the supplementary parameter "--p-maxaccepts 1." PICRUSt2 (version 2.3.0) was used to predict the functional pathways of microorganisms based on ASV characteristic sequences and abundance tables, mainly using Kyoto Encyclopedia of Genes and Genomes (KEGG) and metabolic pathways from all domains of life (MetaCyc) databases (24).

R (version 4.2.2) was utilized for downstream data processing. The Rarefy() function from the GUniFrac package (version 1.8) was used to normalize all samples to a standard of 25,000 to remove the influence of differences in sequencing depth. Spearman correlation was used to calculate the correlation and significance between fat deposition traits and other economic traits of sheep, and the pheatmap package (version 1.0.12) was used to draw the correlation heat map. The vegan package (version 2.6–4) was used to perform principal coordinate analysis (PCoA) of microbial communities based on Bray distance (25), and analysis of similarities (ANOSIM), analysis of variance using distance matrices (ADONIS), and multi-response permutation procedure (MRPP) were used to test the significance of differences between groups. We constructed a random forest regression model using the randomForest package (version 4.7–1.1), with the relative abundances of genera in the cecum greater than 0.1% as feature variables and the BMI trait of sheep as the target variable. The randomForest() function was employed for model construction to identify biomarkers associated with fat deposition in sheep (26), and the replicate() function with the supplementary parameter "cv.fold = 10 step = 1.5" was utilized for cross-validation. We utilized the stat_cor() function based on the Spearman method to construct a linear fitting model to assess the correlation between genera and BMI. When $P < 0.05$, genera were considered to be significantly correlated with BMI, and when $P < 0.1$, genera were considered to have a significant correlation trend with BMI. These genera were defined as key biomarkers. Used the simper() function from the vegan package (version 2.6–4) to perform SIMPER analysis on biomarkers. A correlation network graph was constructed based on Spearman correlation and visualized using Cytoscape (version 3.10.0). All statistical tests in this study were conducted using the Wilcoxon rank sum test, with a significance threshold set at $P < 0.05$. Conducted mediation analysis using Model 4 in the PROCESS (version 4.1) plugin of IBM SPSS Statistics 27, employing the Bootstrap method (5,000 iterations) for mediation effect testing.

## RESULTS

### BMI and fat deposition properties

A total of 303 Hu sheep were used in this cross-sectional study. Descriptive statistics of fat deposition traits are summarized in Table S2. BMI was used as a comprehensive evaluation index for sheep fat deposition traits. Based on BMI, individuals with high (HB) and low (LB) extreme BMI, as well as control BMI (MB), were selected ($n = 60$, 20% of the cohort). The average BMI of the large tested group (303 Hu sheep) was 85.28 kg/m$^2$ (Table S2), and there were significant differences in BMI among the three groups ($P <$ 0.01; Fig. 1a). For other fat deposition traits, there were significant differences between the LB and HB groups, including tail fat, mesenteric fat, total fat, and their corresponding relative weights (body weight; $P < 0.05$; Fig. 1a through c). At the same time, we observed that the area of individual lipid droplets in the mesenteric and perirenal fat of individuals

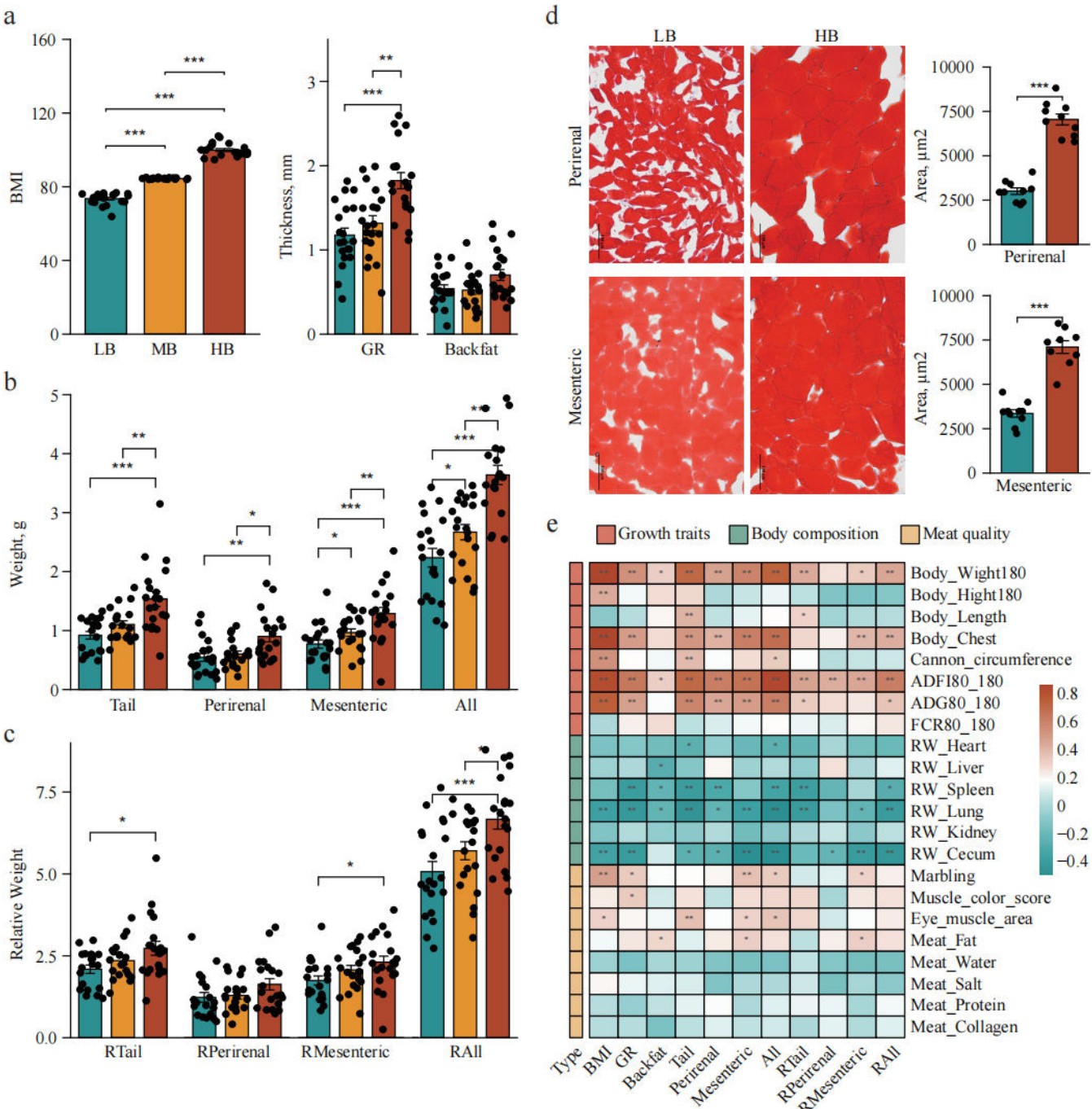

**FIG 1** Analysis of phenotypic trait differences between groups. (a) BMI, carcass fat content value (GR) value, and backfat thickness. (b) Tail fat weight, perirenal fat weight, mesenteric fat weight, and the total weight of the three fats. (c) Relative weight (body weight) of tail fat weight, perirenal fat weight, mesenteric fat weight, and the total weight of the three fats. (d) Oil Red O staining of visceral fat from high and low BMI sheep, and lipid droplet area was calculated. (e) Heat map of the correlation between fat deposition-related traits and sheep growth traits, body composition, and meat quality traits.

in the HB group was significantly higher than that in the LB group ($P < 0.01$; Fig. 1d). These findings indicate that BMI effectively reflects fat deposition ability. Correlation analysis showed that ADFI was significantly positively correlated with all fat deposition traits ($P < 0.01$; Fig. 1e). Marbling and eye muscle area were significantly positively correlated with BMI, indicating that fat deposition plays a positive role in improving meat quality ($P < 0.05$).

## Sheep cecal microbiota and prediction of its functions

We successfully amplified 16S rDNA sequences from sheep cecal digesta samples. After sequencing, 4.33 million raw tags were spliced, and 3.30 million effective tags were obtained by filtering low-quality sequences, short-read sequences, and chimeras for subsequent analysis. The average length of effective tags is 411 bp (Table S3). After denoising by DATA2, 5,296 ASVs common to at least two samples were obtained. As the number of samples and sequencing depth increases, the species accumulation curve and dilution curve tend to be flat, which indicates that the sample size and sequencing depth meet the analysis requirements (Fig. S1). The obtained ASVs were annotated based on the Silva 138 database to describe the composition of the sheep cecal microbiota. The relative abundance of species in the top 10 at different taxonomic levels is displayed. In the cecum, Firmicutes is the main dominant bacterial phylum, with a relative abundance of 64.29%, followed by Bacteroidota (23.70%; Fig. 2a). The rates of Spirochaetota, Verrucomicrobiota, and Desulfobacterota were 2.66%, 2.34%, and 2.08%, respectively. The dominant classes in the cecum include Clostridia (60.71%) and Bacteroidia (23.68%; Fig. 2b). Figure 2c shows the dominant taxa at the order level, including Oscillospirales (29.20%), Bacteroidales (23.50%), and Lachnospirales (14.98%). The dominant bacterial families we identify include Lachnospiraceae (14.90%), Oscillospiraceae (12.35%), Eubacterium_coprostanoligenes (7.5%), Rikenellaceae (7.34%), Prevotellaceae (6.75%), etc. (Fig. 2d). At the genus level, some genera have no specific names. The dominant genera include *UCG-005* of Oscillospiraceae (8.71%), *Rikenella-ceae_RC9_gut_group* (7.50%), *Bacteroides* (4.91%), *Monoglobus* (4.20%), etc. (Fig. 2e). Within the top 10 genera, six belonged to Firmicutes, three to Bacteroidota, and one to Spirochaetota (Fig. 2f).

Functional prediction using PICRUSt2 and KEGG pathway annotation indicated that microbial functions were primarily associated with metabolism, environmental information processing, and cellular processes (Fig. 2g). Sub-pathways involved amino acid metabolism, carbohydrate metabolism, and cofactor and vitamin metabolism.

## Inter-group population diversity analysis

Among the three groups (HB, LB, and MB), 2,651 ASVs were shared, accounting for over 50% of identified ASVs (Fig. 3a). Correlation analysis revealed lower similarity between HB and LB groups compared to MB with either group (Fig. 3b). Alpha diversity showed no significant differences among the three groups (Fig. 3c). Based on the count statistics of ASVs, the Bray-Curtis distance was calculated, and the PCoA was performed (Fig. 3d). There is a clear separation between the LG group and the HB group, with significant differences in PCoA1 and PCoA2 ($P < 0.01$), while the position of MB was between the two groups (Fig. 3d). This shows that there are differences in the structure of the cecal flora between the HB and LB groups, while there are similarities between the MB group and the other two groups, which is consistent with the expected results. We used three test methods to verify the significance of differences in community structure. Three test methods, ANOSIM, ADONIS, and MRPP, were employed to assess community structure differences. They reveal significant differences between the LB and HB groups ($P < 0.01$), whereas differences between MB and the other groups were non-significant ($P > 0.05$; Fig. 3e).

## Differences in cecal microorganisms of sheep with different fat deposits

We observe that at the phylum level, Firmicutes abundance decreased and Bacteroidota abundance increased across the LB, MB, and HB groups (Fig. 3f). The Firmicutes in the LB group are significantly higher than those in the HB group, while the Bacteroidota are significantly lower than those in the HB group ($P < 0.05$; Table S4). The Firmicutes/Bacteroidota value of the LB group was significantly higher than that of the HB group ($P < 0.05$; Table S4). The relative abundance of Lachnospiraceae at the family level shows a decreasing trend, and species composition was similar between groups (Fig. 3g). In order to reduce the bias caused by grouping, a regression analysis was performed on the

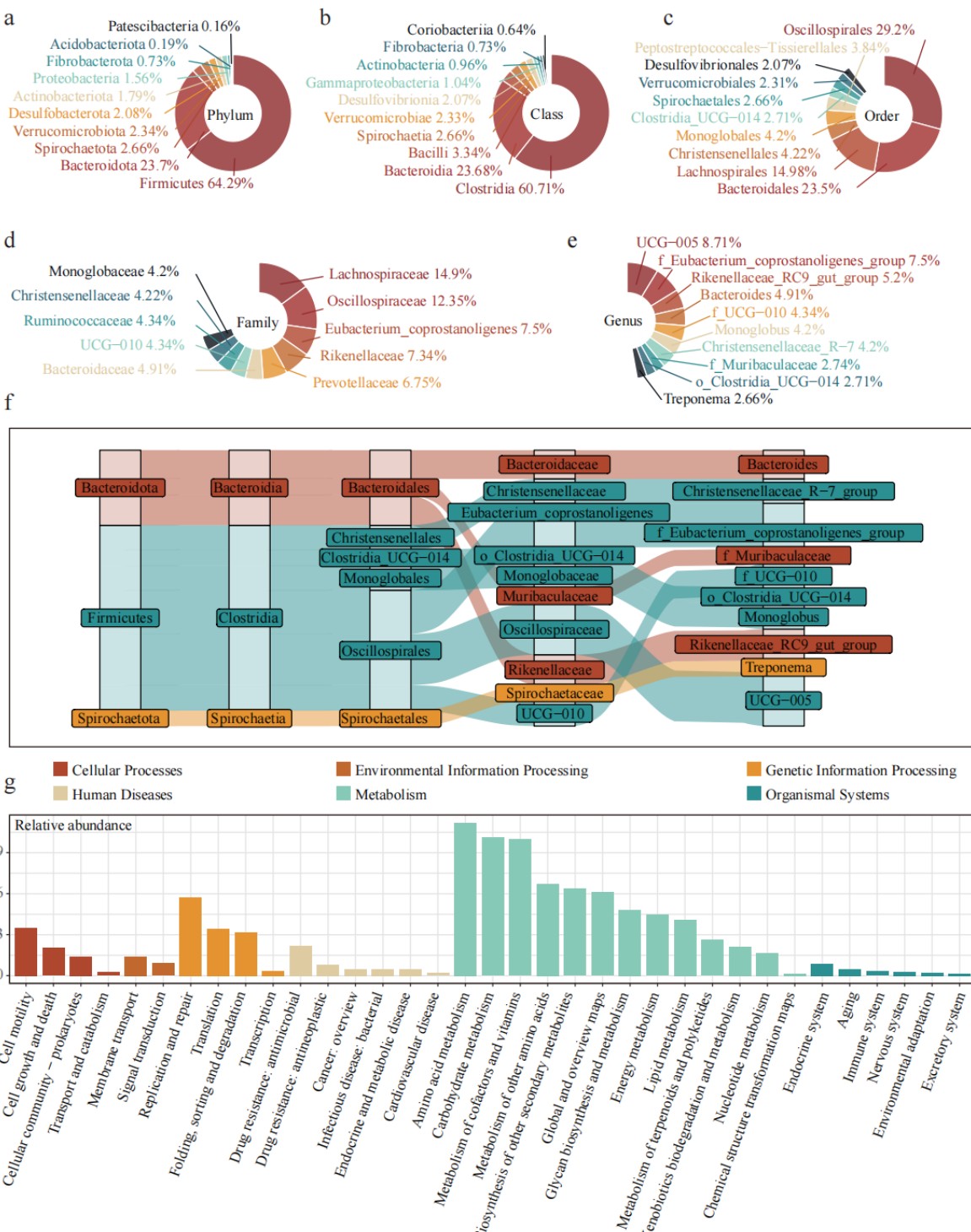

FIG 2 Descriptive analysis of Hu sheep cecal microorganisms. Relative abundance of top 10 species at (a) phylum level, (b) class level, (c) order level, (d) family level, and (e) genus level. (f) Classification of top 10 genera at different species levels. (g) KEGG functional pathway prediction of cecal microorganisms.

relative abundance and BMI of sheep cecal bacteria at the genus level based on the RF machine learning algorithm, and a correlation model between fat deposition and cecal microbiota was constructed (Fig. 3h). We evaluated the model by performing 500 permutation tests, and the results showed that the model was not overfitting (Fig. S2). The cecal microbiota derived from this model explains 27.05% of the variation in fat

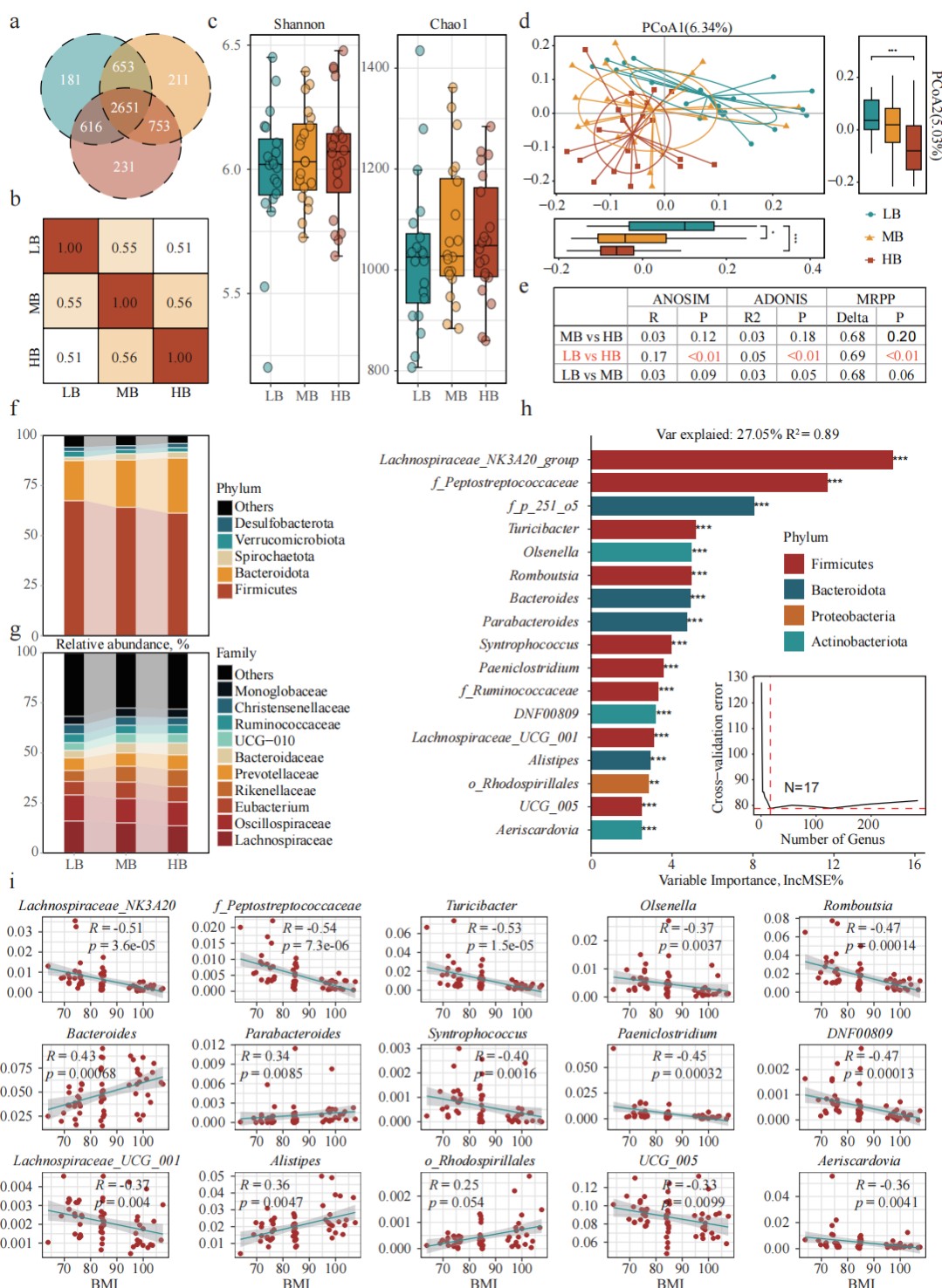

FIG 3 Differences in cecal microorganisms of sheep with different fat deposits. (a) VENN plot of ASVs. (b) Spearman correlation of ASVs between groups. (c) Alpha diversity difference analysis, Shannon, and Chao1 index. (d) PCoA analysis based on Bray distance. The bottom box plot shows the values of PCoA1, while the right box plot shows the values of PCoA2. *, $P < 0.05$; **, $P < 0.01$; ***, $P < 0.001$. (e) Community structure difference testing (ANOSIM, ADONIS, and MRPP). (f) Relative abundance of phylum-level and (g) family-level in different groupings. (h) Biomarkers identified based on RF regression model and 10-fold cross-validation. (i) Linear fitting correlation analysis between biomarkers and BMI.

deposition levels in sheep. To identify the main drivers associated with changes in sheep fat deposition, we used five repeats of 10-fold cross-validation to assess the importance of bacteria. The number of classes in the error curve stabilizes when 17 genera are used,

so we think that these 17 genera may be key biomarkers. Subsequently, these 17 bacterial genera were subjected to linear correlation analysis with BMI. Among them, 14 bacterial genera are significantly correlated with BMI ($P < 0.05$), while the unclassified *o_Rhodospirillales* shows a significant trend ($P < 0.10$; Fig. 3i). Therefore, we define these 15 genera as cecal biomarkers that affect sheep fat deposition, including *Lachnospirilacea_NK3A20_group*, *Turiciactor*, and *Olsenella*.

SIMPER analysis was conducted to assess the contribution of cecal biomarkers to intergroup differences (Fig. S3). The results indicated that both *Bacteroides* and *UCG_005* had contributions exceeding 0.15, suggesting they may be the primary genera responsible for the observed differences. Additionally, *Romboutsia*, *Turicibacter*, and *Lachnospiraceae_NK3A20_group* also demonstrated relatively high contributions. These biomarkers were significantly correlated with fat deposition traits ($P < 0.05$), consistent with linear regression models (Fig. S4).

## Correlation network and differential functions of cecal biomarkers and host traits

We measured the VFAs in the cecum, and the results showed that acetic acid had the highest proportion of VFAs in the cecum, and isobutyric acid in the LB group is significantly higher than in the other two groups ($P < 0.05$; Table S5). At the same time, we also measured the biochemical indicators in the blood, and the TG in the HB group is significantly higher than that in the LB group ($P < 0.05$; Table S5).

We conducted a correlation network analysis to explore the interaction between cecal biomarkers and host fat deposition processes, incorporating cecal microbial metabolites, blood metabolites, and fat deposition traits. The network diagram displays correlation relationships when the Spearman correlation coefficient exceeds 0.3 in absolute value and $P < 0.05$ (Fig. 4a). The results indicate a predominantly positive correlation within the cecal biomarker group. *Lachnospiraceae_NK3A20_group* exhibits a significant positive correlation with *Aeriscardovia*, *Syntrophococcus*, *UCG_005*, *Lachnospiraceae_UCG_001*, *Romboutsia*, and other bacteria ($P < 0.05$). Within the cecal microbial metabolites, acetic acid exhibits a significant negative correlation with valeric acid, propionic acid, and butyrate ($P < 0.05$) and a significant positive correlation with *Lachnospiraceae_NK3A20_group* ($P < 0.05$). Moreover, TG levels were positively correlated with BMI and fat traits ($P < 0.05$), while acetic acid was negatively correlated with TG ($P < 0.05$). Further mediation analysis, as shown in Table S6, demonstrated that the direct effect of the *Lachnospiraceae_NK3A20_group* on BMI was −684.73, with a confidence interval that did not include 0, indicating a significant direct effect. Moreover, the mediating effect value via TG was −114.62, also with a confidence interval excluding 0, suggesting that TG plays a significant mediating role in this mechanism.

Based on the KEGG functional pathways predicted by us, differences between the functional pathways were analyzed. As shown in Fig. 4b, there are nine level 2 pathways that exhibit inter-group differences. Glycan biosynthesis and metabolism, biosynthesis of other secondary metabolites, and cell growth and death in the HG group are significantly higher than those in the LG group ($P < 0.05$). Concurrently, we performed a differential analysis of cecal microbial metabolic pathways using the MetaCyc database (Fig. 4c). There are differences between groups in a total of 24 pathways. The fatty acid elongation -- saturated and gluconeogenesis I pathways in the HG group are significantly higher than those in the LG group ($P < 0.05$).

## DISCUSSION

Adipose tissue is considered an important metabolic organ, playing an important role in the regulation of the body's energy storage and the secretion of related active factors (27). The fat deposition process has obvious local characteristics, mainly including visceral fat, subcutaneous fat, intramuscular fat, etc. In sheep, there is also unique tail fat, which is also a type of subcutaneous fat (28). BMI, as a comprehensive index to evaluate

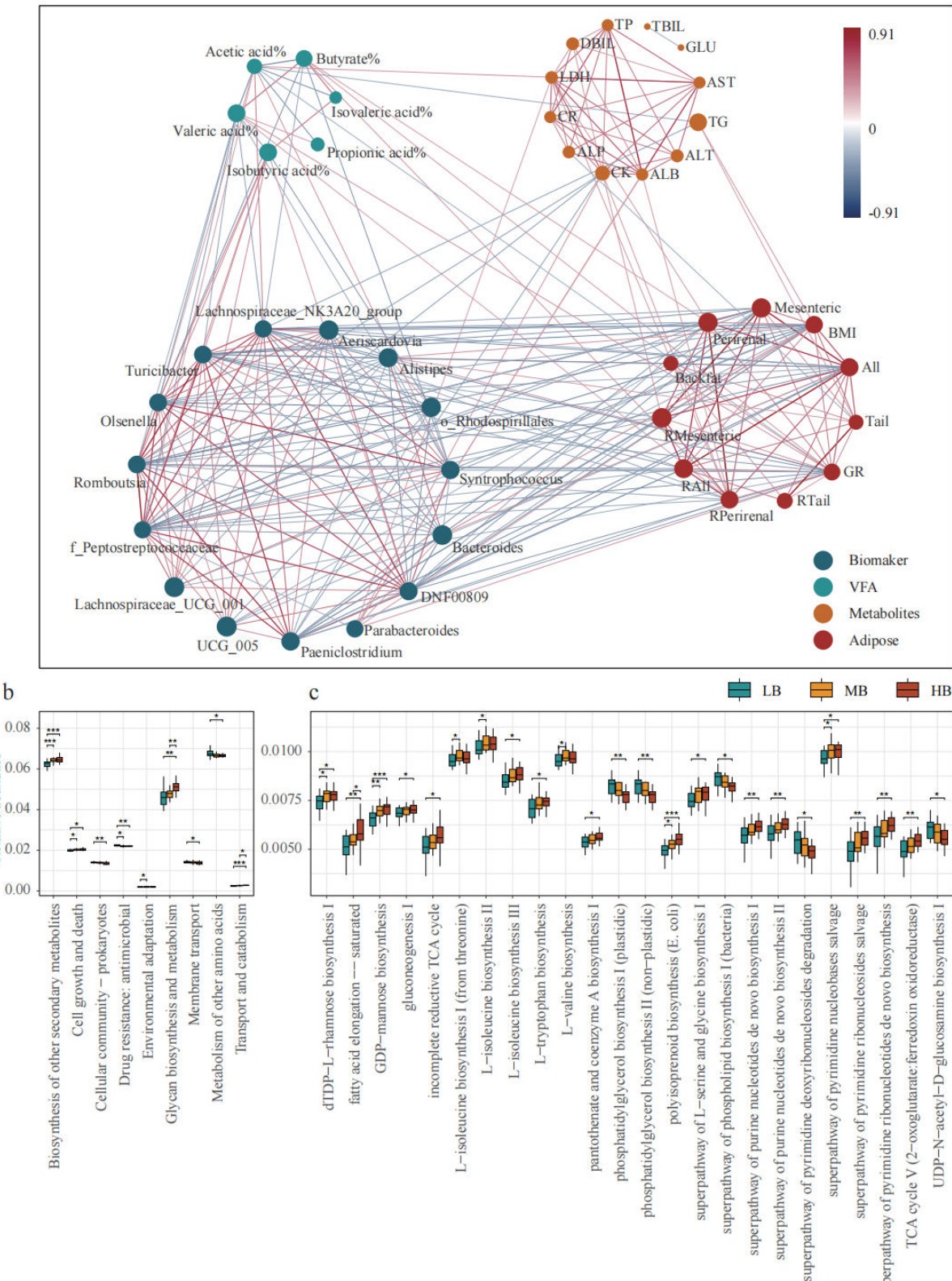

**FIG 4** Analysis of cecal microbial regulatory pathways. (a) The correlation network diagram constructed based on Spearman correlation, including biomarkers, cecal VFAs, blood biochemical indicators, and fat deposition traits. (b) Analysis of differences between groups in KEGG functional pathways. (c) MetaCyc metabolic pathway difference analysis between groups.

fat deposition, is widely used in the evaluation of human obesity, and it has a strong positive correlation with body fat percentage, visceral fat content, and subcutaneous fat content (29–31). There are studies on introducing the BMI index into the study of fat deposition in livestock. Vahedi et al. calculate the BMI of York pigs by measuring body length and weight and use this to evaluate the fat content of the whole body (32). The results of this study show that in different BMI groups, multiple traits related to fat deposition of sheep also had the same difference trend, and BMI had a significant positive correlation with fat deposition-related traits. Therefore, we believe that BMI can be used as an effective comprehensive index to evaluate fat deposition traits of sheep. At the same time, we also observe a significant positive correlation between fat deposition traits and sheep's ADFI and ADG, indicating that higher feed intake can play a positive role in fat deposition and lead to an increase in daily weight gain. A higher growth rate results from a higher appetite, which affects fat deposition (33).

In addition to genetic and epigenetic factors, increasing evidence suggests that digestive tract microbiota is a key factor driving host phenotypic plasticity. Research on the digestive tract of ruminants often focuses on the rumen. Wang et al. reported that rumen flora affects intramuscular fat deposition in sheep through the volatile fatty acid pathway (34). However, based on our earlier research results, it was shown that in the digestive tract of sheep with different levels of fat deposition, the largest difference in bacterial flora structure was in the cecum (35). Therefore, we delved into the relationship between cecal microbes and fat deposition. On average, Firmicutes and Bacteroidetes are dominant in all sheep cecum samples, which is consistent with previous findings in ruminants (36). In the cecum of monogastric animals and poultry, Firmicutes and Bacteroidetes have also been observed as the dominant microbial phyla (37, 38). Both the rumen and cecum belong to the fermentation sites of the digestive tract, and the relative abundance of Bacteroides in the rumen is the highest (12), while Firmicutes in the cecum is the highest, which also reflects that the cecum mainly ferments and utilizes the nutrients that are not fermented in the rumen. *Rikenellaceae_RC9_gut_group* and *Oscillospiraceae_UCG-005*, which have the highest relative abundance at the genus level, are related to lipid metabolism and fatty acid composition (39). The research results of Guerra et al. show that the metabolic function of microorganisms in the gastrointestinal tract of goats is mainly related to carbohydrate, amino acid, and energy metabolism, which is consistent with our research results (40).

The results of the diversity analysis showed that there is no significant difference in the diversity and richness of cecal microorganisms between different BMI groups, but there are significant differences in community structure, which indicates that our grouping based on BMI is effective. There was no significant difference between the MB group and the two extreme groups, suggesting that the microbiota composition of the MB group may reflect a transitional state. Interestingly, we observe that the relative abundance of cecal Firmicutes gradually decreased with increasing BMI, whereas the opposite is observed for Bacteroidetes. In obesity studies in humans, the abundance of Firmicutes increases with increasing BMI, while the abundance of Bacteroidetes gradually decreases (41). Ren et al.'s study on Tibetan sheep shows that the Firmicutes in the rumen of the group with high marbling and intramuscular fat is significantly higher than that of the group with low intramuscular fat (42). Therefore, we believe that the cecum, as the second fermentation site of ruminants, increases the consumption of relevant nutrients when the abundance of Firmicutes in the rumen increases. When chyme moves through the intestine to the cecum, the abundance of Firmicutes decreases due to the reduction of nutrients required by Firmicutes, while the abundance of Bacteroidetes increases (43). Previously reported research results show that the relative abundance of Firmicutes in the rumen of Angus with high RFI is higher than that with low RFI, but the opposite phenomenon occurs in the corresponding cecum, which is consistent with our view (43). The inverse relationship between Firmicutes abundance and abdominal fat is also shown in the cecum of non-ruminants (44).

We further hypothesize that sheep cecal microbial composition influences the ability of sheep to obtain energy from their diet. In order to explore the specific bacterial genera related to fat deposition in the sheep cecal microbiota, we process the data set based on the perspective of regression (RF regression model) and combine the linear correlation model to verify the results. Finally, 15 key genus-level biomarkers are obtained. For ruminants, VFAs produced by digestive tract microorganisms are the main source of energy. The generated VFAs are absorbed into the bloodstream by the rumen or large intestine wall and then used by the host, which contributes about 70% of the heat required by the host (45). Our results show that two bacterial genera of Lachnospiraceae are significantly negatively correlated with sheep fat deposition, among which *Lachnospiraceae_NK3A20_group* is significantly positively correlated with the relative content of acetic acid in the cecum and significantly negatively correlated with TG in the blood. Kaminsky et al. isolated and cultured *Lachnospiraceae_NK3A20_group* and found that when it was co-cultured with methanogen, its main product was acetate (46). The *Lachnospiraceae_NK3A20_group* is abundant in cattle on a high-starch diet and is involved in the fermentation of glucose to lactate, further confirming its positive correlation with acetate levels (47, 48). The study results on goats also indicate a significant positive correlation between the abundance of the *Lachnospiraceae_NK3A20_group* and daily weight gain, which is similar to our findings (49). This suggests that the *Lachnospiraceae_NK3A20_group* may play a similar role in the digestive tract of ruminants. There are more methanogens in the digestive tract of ruminants, so the metabolite of *Lachnospiraceae_NK3A20_group* in the cecum was mainly acetic acid. Acetate can reduce the accumulation of TG by inhibiting the production of uric acid and lactic acid (50). Therefore, we believe that acetic acid in the cecum may affect the TG concentration in the blood after entering the blood through the intestinal wall. Moreover, we found that acetic acid is significantly positively correlated with LDH, which may be related to inhibiting the concentration of lactic acid. The research results of Qiu et al. showed that the bacterial genus *Lachnospiraceae* belongs to was significantly negatively correlated with TG, which is similar to our results (51). TG is transported to adipose tissue through blood circulation and stored there (52). Many studies show that the TG content in the blood is significantly positively correlated with fat deposition-related traits (53). These results indicate that the *Lachnospiraceae_NK3A20_group* in the cecum may affect the fat deposition process in sheep through the VFAs-TG pathway. Our previous research results indicate that acetate produced by microorganisms in the digestive tract of Hu sheep, Tan sheep, and Dorper sheep is negatively correlated with muscle fat deposition, which is consistent with the findings of this study (12). Research results also showed that microorganisms might have regulated the host's metabolism through the gut-brain axis, with the gut microbiota influencing the bidirectional communication between the gut and the brain (54). Previous research results showed that intestinal flora and their metabolites could lead to obesity (55). It was also related to the strong energy acquisition ability of gut microorganisms associated with obesity (56). At the same time, there might have been a process of bacterial cross-feeding between positively correlated biomarkers, where one microbial community provided nutrients for another (57). The research results on mice showed that some gut microorganisms were significantly negatively correlated with body weight and serum TG, which was similar to our results (58).

In previous reports, *Turicibacter* is identified as a regulator of host adipobiology, colonization in the host effectively reduces serum triglycerides and adipose tissue mass (59), and *Turicibacter* shows a significant positive correlation with *Lachnospiraceae_NK3A20_group*. Moreover, the abundance of *Turicibacter* also showed a gradual increase at the nine developmental time points of sheep from 0 to 180 days (60). This study shows that 11 biomarkers that are significantly negatively correlated with fat deposition were significantly positively correlated with each other, indicating that they may have played a synergistic role. There are fewer biomarkers that are positively correlated with fat deposition traits, including *Bacteroides*, *Parabacteroides*, and *Alistipes*,

which is similar to the results of previous studies (61, 62). Among them, *Parabacteroides* has a significant positive correlation with ALB and CK in serum, which may reflect that it may indirectly participate in protein and energy metabolism, thereby affecting the levels of albumin and creatine kinase in serum. When Wouters et al. fed mice a high-fat diet, *Parabacteroides* was highly positively correlated with fat intake, and it may be involved in the regulation of the brain-gut axis (63). Our preliminary baseline study results indicated that when using sheep feces for microbiota transplantation in mice, the abundance of *Bacteroides* significantly increased. Furthermore, recipient mice that received high-fat sheep feces had higher body weight and relative abundance of *Bacteroides* compared to those that received low-fat sheep feces, which is consistent with our findings in sheep (64).

Differential functional pathways also give us some hints. Biosynthesis of other secondary metabolites and glycan biosynthesis and metabolism in the HB group are significantly higher than those in the LB group, suggesting that the metabolic rate of the cecal microorganisms in the HB group is faster, and the production of metabolites is higher. MetaCyc database results show that the HB group is significantly higher than the LB group in metabolic pathways such as fatty acid synthesis and amino acid synthesis, which is consistent with our expectations.

The results of this study suggest that modulating the cecal microbiota has significant potential to improve livestock leanness. In practical applications, incorporating high-fiber feed could selectively promote specific microbial taxa associated with lean tissue deposition, thereby enhancing livestock productivity (65). Furthermore, previous studies have shown that probiotic supplementation significantly increases the abundance of Lachnospiraceae (66), which highlights the potential value of *Lachnospiraceae_NK3A20_group* as a promising target for further research. Furthermore, previous studies have demonstrated that the hindgut microbiota in ruminants plays a critical role in feed fermentation and feed efficiency (60, 67). Our study further reveals a strong association between hindgut microbiota and host fat deposition. This not only confirms the essential role of hindgut microbiota in host metabolism but also provides a novel perspective for optimizing livestock leanness through the modulation of cecal microbiota.

Overall, this study contributes to advancing our understanding of the regulatory pathways underlying fat deposition traits in sheep, which could accelerate the development of efficient microbial additives. Despite these findings, certain limitations should be acknowledged. While we have identified some potential biomarkers and explored possible mechanisms, further functional validation, such as single-strain isolation and culture or microbiota transplantation, is still required.

## Conclusion

In summary, our findings elucidate the structural and functional aspects of the cecal microbiome in Hu sheep, revealing correlations between the cecal microorganisms and host fat deposition traits and identifying key bacterial groups driving these associations. Distinct differences in the bacterial community structure are observed between the low- and high-fat deposition groups. Through regression models and linear correlation analyses, we identify 15 genus-level biomarkers. Our results suggest that these cecal biomarkers may influence the host's fat deposition process via a pathway involving the cecal microbiota, cecal VFAs, blood TG, and fat. These findings indicate that fat deposition could potentially be modulated by developing nutritional strategies aimed at regulating the cecal microbiome composition. Further research is needed to validate these findings and explore the practical applications of such strategies in the context of animal production and health.

## ACKNOWLEDGMENTS

This research was funded by the National Natural Science Foundation of China (32372850, 32260818 and 3247200928), the Gansu Provincial Major Science and Technology Project (22ZD6NC069) and the Gansu Provincial Natural Science Foundation Excellent Doctoral Student Project (25JRRA732).

## AUTHOR AFFILIATIONS

[1]The State Key Laboratory of Grassland Agro-ecosystems, College of Pastoral Agriculture Science and Technology, Lanzhou University, Lanzhou, Gansu, China

[2]College of Animal Science and Technology, Gansu Agricultural University, Lanzhou, Gansu, China

## AUTHOR ORCIDs

Jiangbo Cheng  http://orcid.org/0000-0002-1281-5157
Xiaobin Yang  http://orcid.org/0000-0002-8475-1287
Weimin Wang  http://orcid.org/0000-0002-6660-4865

## FUNDING

| Funder | Grant(s) | Author(s) |
| --- | --- | --- |
| National Natural Science Foundation of China | 32372850 | Weimin Wang |
| National Natural Science Foundation of China | 32260818 | Weimin Wang |
| Gansu Provincial Major Science and Technology project | 22ZD6NC069 | Weimin Wang |

National Natural Science Foundation of China (3247200928)Gansu Provincial Natural Science Foundation Excellent Doctoral Student Project (25JRRA732)

## DATA AVAILABILITY

The raw sequencing data generated in this study have been uploaded to the NCBI Sequence Read Archive (submission ID: SUB14387386 and BioProject ID: PRJNA1101910).

## ETHICS APPROVAL

This study complies with the regulations and guidelines of the Gansu Provincial People's Congress Government and was conducted in strict accordance with animal care and experimental procedures. The study was approved by the Animal Welfare and Ethics Committee of Lanzhou University (No.: 2020-01 and 2021-02).

## ADDITIONAL FILES

The following material is available online.

### Supplemental Material

**Supplemental material (Spectrum01488-24-s0001.docx).** Fig. S1 to S4; Tables S1 to S6.

### Open Peer Review

**PEER REVIEW HISTORY (review-history.pdf).** An accounting of the reviewer comments and feedback.

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
