## [Reviewer comments · Microbiology Spectrum]

Microbiology Spectrum

Exploring the cecal microbial community associated with fat deposition in sheep and its possible pathways of action

Jiangbo Cheng, Dan Xu, Deyin Zhang, Kai Huang, Yukun Zhang, Xiaolong Li, Yuan Zhao, Liming Zhao, Quanzhong Xu, Xiaobin Yang, Zongwu Ma, Huibin Tian, Xiaoxue Zhang, and Weimin Wang

Corresponding Author(s): Weimin Wang, Lanzhou University

Review Timeline:

Submission Date:	June 17, 2024
Editorial Decision:	September 25, 2024
Revision Received:	October 28, 2024
Editorial Decision:	January 18, 2025
Revision Received:	February 27, 2025
Editorial Decision:	March 16, 2025
Revision Received:	April 8, 2025
Accepted:	April 14, 2025

Editor: Bernadette Connors

Reviewer(s): Disclosure of reviewer identity is with reference to reviewer comments included in decision letter(s). The following individuals involved in review of your submission have agreed to reveal their identity: Jian Zhou (Reviewer #2)

Transaction Report:

DOI: <https://doi.org/10.1128/spectrum.01488-24>

Re: Spectrum01488-24 (Exploring the cecal microbial community associated with fat deposition in Sheep and its possible pathways of action)

Dear Dr. Weimin Wang:

Thank you for the privilege of reviewing your work. Below you will find my comments, instructions from the Spectrum editorial office, and the reviewer comments.

Revision Guidelines

Sincerely,
Bernadette Connors
Editor
Microbiology Spectrum

Reviewer #2 (Comments for the Author):

The paper titled "Exploring the cecal microbial community associated with fat deposition in Sheep and its possible pathways of action" investigates the relationship between cecal microbiota and fat deposition in sheep, focusing on Hu sheep. The study employs 16S rDNA sequencing to identify key microbial biomarkers associated with fat deposition and explores the potential metabolic pathways through which these microbes may influence fat accumulation. The findings suggest that the cecal microbiota plays a significant role in fat deposition, potentially offering insights for improving sheep production efficiency through

microbiome management.

Paper Strengths

Innovative Methodology: The paper employs a comprehensive approach combining 16S rDNA sequencing, random forest regression, and linear correlation analyses to identify key microbial biomarkers related to fat deposition. This methodology is innovative and provides a detailed understanding of the microbial communities in the cecum and their potential functional roles.

Significant Findings: The study's identification of specific bacterial genera that correlate with fat deposition traits in sheep, such as the *Lachnospiraceae_NK3A20_group* and their association with acetic acid and triglyceride levels, is particularly insightful. These findings open new avenues for research in animal husbandry and microbiome manipulation.

Practical Implications: The results have practical implications for sheep breeding and production. By understanding the microbiota's role in fat deposition, there is potential to develop strategies to control or optimize this process, improving meat quality and production efficiency in the industry.

Paper Weaknesses

Limited Sample Size: The study uses a relatively small sample size (n=60), which may limit the generalizability of the findings. A larger sample size would provide more robust results and increase the confidence in the identified biomarkers.

Lack of Functional Validation: While the study identifies potential pathways and correlations, it lacks experimental validation of these pathways. Functional studies to directly assess the role of identified microbes in fat deposition would strengthen the conclusions.

Inadequate Discussion on Broader Implications: The paper could benefit from a more in-depth discussion on the broader implications of its findings, particularly in relation to other ruminants or livestock species. Comparisons with similar studies in different animals could provide additional context and relevance.

Questions to Authors and Suggestions for Rebuttal

Can the authors provide more details on the selection criteria for the specific bacterial genera that were identified as biomarkers? How were the thresholds for significance determined?

The study mentions correlations between microbial metabolites and fat deposition traits, but how consistent are these correlations across different sheep populations or breeds? Have similar studies been conducted in other ruminant species?

Could the authors suggest potential interventions or treatments that could be derived from their findings? For example, could dietary modifications be used to influence the cecal microbiota and, in turn, fat deposition?

Overall Score: 7/10

The paper presents a well-conceived and executed study with significant findings that contribute to our understanding of the cecal microbiota's role in fat deposition in sheep. However, the limitations in sample size and lack of functional validation prevent it from achieving a higher score. Addressing these weaknesses in future research could significantly enhance the impact and applicability of the study.

Reviewer #3 (Comments for the Author):

General comments:

English language needs to be revised.

Whole paper is a mixture of past and present tenses making it confusing and difficult to understand.

Capitalization of words within the sentences should also be corrected.

Genera name are not italic at some place.

Specific comments:

Abstract: It is better to state results obtained instead of listing the names of the analysis used. For examples, which VFA were significant? Which bacteria were abundant?

Importance: need improvement. It is not clear how fat deposition is associated with increased production efficiency? Why cecal microbiota are associated with fat deposition and not rumen or intestinal?

Line 56: how fat devalues meat? Is it associated with consumer's demand? Please add more solid argument.

Line 64: In the digestive tract of monogastric animals, the cecum is the most critical fermentation site due to high bacterial diversity, making it a primary focus of research.

Line 66: remove word unique

Line 77-79: please rewrite the sentence.

Line 86: 303 hu sheep? In abstract you mentioned 60 sheeps.

Material and methods

Line 107: Body mass index, average daily.....was calculated.

Line 111: 20% of the population was used....

Line 112: BMI was used to sort the population and 20 sheep were.....were selected and 20 sheep at the median were taken as

.....

Line 119: complete cecum was obtained.....samples were collected and stored.....

Please correct the description of method in the same way mentioned above.

Line 189: how do you define BMI in three groups? Was it an average value?

Line 198: did you measure body composition and meat quality? If you are using data from previously published paper then cite it and mention it clearly in the methodology section.

Line 201-202: How correlation between these two parameters is enough to indicate their importance in meat quality?

Line 205: raw tags

Line236: inter-group population.....

Line 255-259: are the changes in the relative abundance in fig 3e are significant? You have already showed the abundance of taxa in fig 2.

Line270-271: what do you mean? P value in the figure is non-significant.

Line 351: reference?

Discussion

References should be added to support the argument.

Figure legends

Line 597: ANOSIM

Response to Reviewer 2 Comments

The paper titled "Exploring the cecal microbial community associated with fat deposition in Sheep and its possible pathways of action" investigates the relationship between cecal microbiota and fat deposition in sheep, focusing on Hu sheep. The study employs 16S rDNA sequencing to identify key microbial biomarkers associated with fat deposition and explores the potential metabolic pathways through which these microbes may influence fat accumulation. The findings suggest that the cecal microbiota plays a significant role in fat deposition, potentially offering insights for improving sheep production efficiency through microbiome management.

Paper Strengths

Innovative Methodology: The paper employs a comprehensive approach combining 16S rDNA sequencing, random forest regression, and linear correlation analyses to identify key microbial biomarkers related to fat deposition. This methodology is innovative and provides a detailed understanding of the microbial communities in the cecum and their potential functional roles.

Significant Findings: The study's identification of specific bacterial genera that correlate with fat deposition traits in sheep, such as the *Lachnospiraceae_NK3A20_group* and their association with acetic acid and triglyceride levels, is particularly insightful. These findings open new avenues for research in animal husbandry and microbiome manipulation.

Practical Implications: The results have practical implications for sheep breeding and production. By understanding the microbiota's role in fat deposition, there is potential to develop strategies to control or optimize this process, improving meat quality and production efficiency in the industry.

Paper Weaknesses

Limited Sample Size: The study uses a relatively small sample size ($n=60$), which may limit the generalizability of the findings. A larger sample size would provide more robust results and increase the confidence in the identified biomarkers.

Response 1: We gratefully appreciate for your valuable suggestion. I apologize for the relatively small sample size ($n=60$) used in this study. In this research, we aimed to explore the relationship between fat deposition in sheep and cecal microbiota. Therefore, we selected extreme phenotypic groups based on fat deposition traits from a larger sheep population, including both high and low trait extremes, as well as a control group for in-depth study. Due to limitations in sample collection, we ensured that all sheep were raised in the same environment and that samples were collected at the same age. As a result, we selected 20% of the entire population as the sample size for the study. In future research, we will validate the results within the larger population.

Lack of Functional Validation: While the study identifies potential pathways and correlations, it lacks experimental validation of these pathways. Functional studies to directly assess the role of identified microbes in fat deposition would strengthen the conclusions.

Response 2: We gratefully appreciate for your valuable suggestion. I apologize for the lack of experimental validation of the identified microorganisms in our study. It is indeed essential to conduct functional research that directly assesses the role of identified microorganisms in fat deposition, and your suggestions provide direction for our future research. Currently, we have screened key microorganisms at the genus level based on 16S rDNA sequencing and explored potential pathways of influence through volatile fatty acids and serum metabolites. Additionally, we have supplemented the discussion section with a review of previous reports on the current research status of key microorganisms. In our next phase of research, we will utilize metagenomic sequencing to pinpoint key microorganisms at the species level and perform functional validation of these key strains through cultureomics.

Inadequate Discussion on Broader Implications: The paper could benefit from a more in-depth discussion on the broader implications of its findings, particularly in relation to other ruminants or livestock species. Comparisons with similar studies in different animals could provide additional context and relevance.

Response 3: We gratefully appreciate for your valuable suggestion. A comparison of similar studies across different species has been added to the discussion section.

Questions to Authors and Suggestions for Rebuttal

Can the authors provide more details on the selection criteria for the specific bacterial genera that were identified as biomarkers? How were the thresholds for significance determined?

Response 4: We gratefully appreciate for your valuable suggestion. In the process of identifying biomarkers, we first used a random forest model to determine the contribution of each genus to BMI variation, identifying those with significant variable importance ($P < 0.05$). Subsequently, we applied five repeats of 10-fold cross-validation to assess the importance of these bacteria, defining the number of genera before the elbow point of the cross-validation error as the primary drivers related to changes in sheep fat deposition. We selected 17 genus, which were linearly fitted to BMI traits to validate their correlation with fat deposition. Genera were considered significantly correlated with a $P < 0.05$, and as having a trend towards significance with a $P < 0.1$. Ultimately, we identified 15 biomarkers. This has been supplemented in the Materials and Methods section.

The study mentions correlations between microbial metabolites and fat deposition traits, but how consistent are these correlations across different sheep populations or breeds? Have similar studies been conducted in other ruminant species?

Response 5: We gratefully appreciate for your valuable suggestion. Our previous research results indicated that acetate produced by microorganisms in the digestive tract of Hu sheep, Tan sheep, and

Dorper sheep is negatively correlated with muscle fat deposition, which is consistent with the findings of this study¹. In other ruminants, research has primarily focused on the relationship between microbial metabolites and feed efficiency traits, showing that cattle with higher feed efficiency have higher concentrations of acetate². Our results demonstrate that feed efficiency is negatively correlated with total fat deposition (as FCR is inversely proportional to feed efficiency) and positively correlated with acetate, aligning with previous reports. This suggests that our findings can be applied to other breeds of sheep and ruminants. This has been supplemented in the Discussion section.

1. Cheng J, Zhang X, Xu D, Zhang D, Zhang Y, Song Q, Li X, Zhao Y, Zhao L, Li W, Wang J, Zhou B, Lin C, Yang X, Zhai R, Cui P, Zeng X, Huang Y, Ma Z, Liu J, Wang W. Relationship between rumen microbial differences and traits among Hu sheep, Tan sheep, and Dorper sheep. *J Anim Sci.* 2022 Sep 1;100(9):skac261. doi: 10.1093/jas/skac261.
2. Lage CFA, Coelho SG, Diniz Neto HC, Malacco VMR, Rodrigues JPP, Sacramento JP, Teixeira VA, Machado FS, Pereira LGR, Tomich TR, Campos MM. Relationship between feed efficiency indexes and thermography, blood, and ruminal parameters in pre-weaning dairy heifers. *PLoS One.* 2020 Jul 15;15(7):e0236118. doi: 10.1371/journal.pone.0236118.

Could the authors suggest potential interventions or treatments that could be derived from their findings? For example, could dietary modifications be used to influence the cecal microbiota and, in turn, fat deposition?

Response 6: We gratefully appreciate for your valuable suggestion. In this study, we investigated the relationship between cecal microorganisms and sheep fat deposition traits by linking microorganisms, microbial metabolites, host metabolites, and the host. In future research, we plan to regulate sheep phenotypic traits by modulating the gut microbiota. Our previous findings showed that the cecal and fecal microbiota have similar organizational structures¹, and fecal microbiota transplantation can effectively alter the recipient's gut microbiota and impact host phenotypic traits². Therefore, we will utilize culturomics to identify key microbial strains and conduct experimental validation, providing resources for the development of efficient microbial formulations.

1. Cheng J, Wang W, Zhang D, Zhang Y, Song Q, Li X, Zhao Y, Xu D, Zhao L, Li W, Wang J, Zhou B, Lin C, Zhang X. Distribution and Difference of Gastrointestinal Flora in Sheep with Different Body Mass Index. *Animals (Basel).* 2022 Mar 30;12(7):880. doi: 10.3390/ani12070880.
2. Cheng J, Zhang X, Zhang D, Zhang Y, Li X, Zhao Y, Xu D, Zhao L, Li W, Wang J, Zhou B, Lin C, Yang X, Zhai R, Cui P, Zeng X, Huang Y, Ma Z, Liu J, Wang W. Sheep fecal transplantation affects growth performance in mouse models by altering gut microbiota. *J Anim Sci.* 2022 Nov 1;100(11):skac303. doi: 10.1093/jas/skac303.

Overall Score: 7/10

The paper presents a well-conceived and executed study with significant findings that contribute to our understanding of the cecal microbiota's role in fat deposition in sheep. However, the limitations in sample size and lack of functional validation prevent it from achieving a higher score. Addressing these weaknesses in future research could significantly enhance the impact and applicability of the study.

Response 7: We gratefully appreciate for your valuable suggestion. Thank you for your review of the paper. Your suggestions are of great help to our subsequent research. In the next step of research, we will expand the experimental population, use metagenomic technology to locate the biomarkers to the species level, and verify the function of the biomarkers, in order to provide new ideas for the regulation of sheep traits.

Great respect to you again, thank you for your advice, best wishes to you.

Response to Reviewer 3 Comments

General comments:

English language needs to be revised.

Response 1: We gratefully appreciate for your valuable suggestion. We are very sorry for the language problem and have modified the language expression.

Whole paper is a mixture of past and present tenses making it confusing and difficult to understand.

Response 2: We gratefully appreciate for your valuable suggestion. The tense has been revised throughout the article.

Capitalization of words within the sentences should also be corrected.

Response 3: We gratefully appreciate for your valuable suggestion. Corrections have been made to incorrect capitalization throughout the text.

Genera name are not italic at some place.

Response 4: We gratefully appreciate for your valuable suggestion. Changed genus names to italics.

Specific comments:

Abstract: It is better to state results obtained instead of listing the names of the analysis used. For examples, which VFA were significant? Which bacteria were abundant?

Response 5: We gratefully appreciate for your valuable suggestion. The Abstract section has been supplemented and revised to "Fat deposition is a crucial economic trait during sheep growth and development, closely linked to economic returns. Current research on sheep's digestive tract predominantly focuses on the rumen, the composition of the cecal microbiota and its relationship with host fat deposition remains largely unexplored. In this study, we sequenced the cecal microbiota of 60 Hu sheep, exhibiting marked differences in traits. The most abundant species in the sheep cecum were Firmicutes and Bacteroidota. Statistical analysis results indicate that there are significant differences in microbial community structures among different groups ($P < 0.05$). Employing a random forest regression model and linear regression, we identified 15 biomarkers, including *Lachnospiraceae_NK3A20_group*, *Turcibacter*, and *Bacteroides*, which likely play key roles in varying fat deposition. Additionally, we measured volatile fatty acids (VFAs) in the cecum and biochemical indices in serum. The proportion of acetic acid in the cecal VFAs is the highest, and the isobutyric acid level in the low-fat group of sheep is higher than that in the other groups ($P < 0.05$). The TG level in the high-fat group of sheep is significantly higher than that in the low-fat group ($P < 0.05$). Spearman correlation analysis indicated a strong positive correlation between the *Lachnospiraceae_NK3A20_group* and acetic acid levels, and between serum triglycerides (TG) and fat deposition traits ($P < 0.05$). Conversely, a significant negative correlation was observed between TG

and acetic acid levels ($P < 0.05$). These findings suggest that the cecal microbiota influences fat deposition in sheep, potentially via the VFAs-TG metabolic pathway."

Importance: need improvement. It is not clear how fat deposition is associated with increased production efficiency? Why cecal microbiota are associated with fat deposition and not rumen or intestinal?

Response 6: We gratefully appreciate for your valuable suggestion. Changed Importance section to "Compared with muscle development, fat deposition consumes more energy, and controlling the fat deposition process can effectively reduce energy waste. Current research on rumination mainly focuses on the rumen, but lacks research on the hindgut. The results showed that there were differences in the cecal flora of sheep with different fat deposition levels, and the differential microorganisms were significantly correlated with the metabolite content in the cecum and the host. Therefore, the regulation of cecal microorganisms can help improve the fat deposition characteristics of sheep."

Line 56: how fat devalues meat? Is it associated with consumer's demand? Please add more solid argument.

Response 7: We gratefully appreciate for your valuable suggestion. I am very sorry that the previous expression is inadequate. Since high fat intake causes more metabolic diseases in the human body, the demand for lean meat products in the consumer market increases. Therefore, excessive fat deposition affects the value of meat products. Additionally, fat deposition consumes more energy and increases breeding costs. This part is revised to "Furthermore, reducing fat increases effectively accelerates muscle growth, reduces feed costs, and increases production income¹. At the same time, studies show that meat products with excessive fat content in the human body cause obesity, diabetes, and other related metabolic diseases, which seriously endanger human health. Therefore, the demand for lean products in the consumer market increases^{2,3}". Relevant references are added.

1. Prache S, Schreurs N, Guillier L. Review: Factors affecting sheep carcass and meat quality attributes. *Animal*. 2022 Feb;16 Suppl 1:100330. doi: 10.1016/j.animal.2021.100330.
2. Fuster JJ, Ouchi N, Gokce N, Walsh K. Obesity-Induced Changes in Adipose Tissue Microenvironment and Their Impact on Cardiovascular Disease. *Circ Res*. 2016 May 27;118(11):1786-807. doi: 10.1161/CIRCRESAHA.115.306885.
3. Longo M, Zatterale F, Naderi J, Parrillo L, Formisano P, Raciti GA, Beguinot F, Miele C. Adipose Tissue Dysfunction as Determinant of Obesity-Associated Metabolic Complications. *Int J Mol Sci*. 2019 May 13;20(9):2358. doi: 10.3390/ijms20092358.

Line 64: In the digestive tract of monogastric animals, the cecum is the most critical fermentation site due to high bacterial diversity, making it a primary focus of research.

Response 8: We gratefully appreciate for your valuable suggestion. Changed "In the digestive tract of monogastric animals, the cecum is the most critical fermentation site, has high bacterial diversity, and is the main research object." to "In the digestive tract of monogastric animals, the cecum is the most critical fermentation site due to high bacterial diversity, making it a primary focus of research."

Line 66: remove word unique

Response 9: We gratefully appreciate for your valuable suggestion. The "unique" has been deleted.

Line 77-79: please rewrite the sentence.

Response 10: We gratefully appreciate for your valuable suggestion. Changed "At present, because feeding grain pellet feed can improve the production efficiency of sheep, grain pellet feed is used in large-scale breeding." to "Currently, grain pellet feed is used in large-scale sheep breeding because it improves production efficiency.".

Line 86: 303 hu sheep? In abstract you mentioned 60 sheeps.

Response 11: We gratefully appreciate for your valuable suggestion. In this study, fat deposition phenotypic traits were measured in a large population of 303 Hu sheep. An extreme phenotypic group based on fat deposition traits (n=60) was selected, accounting for 20% of the total population, including a high trait extreme group and a low trait extreme group, as well as a control group for in-depth research.

Material and methods

Line 107: Body mass index, average daily.....was calculated.

Response 12: We gratefully appreciate for your valuable suggestion. Changed "And calculate body mass index (BMI), average daily feed intake (ADFI), average daily gain (ADG), and feed conversion rate (FCR)." to "Body mass index (BMI), average daily feed intake (ADFI), average daily gain (ADG), and feed conversion rate (FCR) were calculated.".

Line 111: 20% of the population was used....

Response 13: We gratefully appreciate for your valuable suggestion. Changed "Use 20% of the population as the test population size (60) for subsequent experiments." to "20% of the population was used as the test population size (60) for subsequent experiments.".

Line 112: BMI was used to sort the population and 20 sheep were.....were selected and 20 sheep at the median were taken as

Response 14: We gratefully appreciate for your valuable suggestion. Changed "Use BMI to sort the population, select 20 sheep at the high and low extremes, and take 20 sheep at the median as the control group." to "BMI was used to sort the population and 20 sheep were selected at the high and low extremes and 20 sheep at the median were taken as the control group.".

Line 119: complete cecum was obtained.....samples were collected and stored.....

Response 15: We gratefully appreciate for your valuable suggestion. Changed "Obtain the complete cecum of the experimental animals, collect cecal chyme samples at the same location, and store the chyme in a -80°C ultra-low temperature refrigerator." to "The complete cecum of the experimental animals was obtained, cecal chyme samples were collected at the same location, and the chyme was stored in a -80°C ultra-low temperature refrigerator.".

Please correct the description of method in the same way mentioned above.

Response 16: We gratefully appreciate for your valuable suggestion. The description in the Methods section has been modified.

Line 189: how do you define BMI in three groups? Was it an average value?

Response 17: We gratefully appreciate for your valuable suggestion. The results section describes the mean BMI value in the 303 large groups (85.28 kg/m²), as shown in Supplementary Table S2. And the BMI of different groups was tested using the Wilcoxon rank sum test.

Line 198: did you measure body composition and meat quality? If you are using data from previously published paper then cite it and mention it clearly in the methodology section.

Response 18: We gratefully appreciate for your valuable suggestion. In this study, the body composition and meat quality of the experimental sheep were measured, and the specific measurement methods are supplemented in the Materials and Methods section.

Line 201-202: How correlation between these two parameters is enough to indicate their importance in meat quality?

Response 19: We gratefully appreciate for your valuable suggestion. Eye muscle area refers to the cross-sectional area of the longissimus dorsi muscle of livestock, which is related to the meat production performance of livestock. The muscle marbling score reflects the deposition of intramuscular fat, which is an important characteristic of meat. It can strongly affect the flavor, tenderness and juiciness of meat¹. Therefore, the significant correlation between BMI and these two indicators shows the correlation between BMI and meat quality.

1. Hocquette JF, Gondret F, Baéza E, Médale F, Jurie C, Pethick DW. Intramuscular fat content in meat-producing animals: development, genetic and nutritional control, and identification of putative markers. *Animal*. 2010 Feb;4(2):303-19. doi: 10.1017/S1751731109991091.

Line 205: raw tags

Response 20: We gratefully appreciate for your valuable suggestion. Changed "Row Tags" to "raw tags".

Line236: inter-group population.....

Response 21: We gratefully appreciate for your valuable suggestion. Changed "Between-group population diversity analysis" to "Inter-group population diversity analysis".

Line 255-259: are the changes in the relative abundance in fig 3e are significant? You have already showed the abundance of taxa in fig 2.

Response 22: We gratefully appreciate for your valuable suggestion. The significant difference analysis between the groups is supplemented (Supplementary Table S4), and the results show that the Firmicutes in the LB group are significantly higher than those in the HB group ($P < 0.05$), while the Bacteroidetes are significantly lower than those in the HB group ($P < 0.05$). Figure 2 shows the overall microbial annotation results of the cecum, while Figure 3e aims to show the dynamic changes in the relative abundance of microorganisms between different groups.

Line270-271: what do you mean? P value in the figure is non-significant.

Response 23: We gratefully appreciate for your valuable suggestion. The results of linear fitting analysis showed that the P value of the unclassified *o_Rhodospirillales* was 0.054, which was between 0.1 and 0.05, so it was considered to have a significant trend.

Line 351: reference?

Response 24: We gratefully appreciate for your valuable suggestion. Relevant reference "The impact of feed efficiency selection on the ruminal, cecal, and fecal microbiomes of Angus steers from a commercial feedlot" has been added.

Discussion

References should be added to support the argument.

Response 25: We gratefully appreciate for your valuable suggestion. Similar findings on other species have been added to the Discussion section to support the arguments of this paper.

Figure legends

Line 597: ANOSIM

Response 26: We gratefully appreciate for your valuable suggestion. Corrected to "ANOSIM".

Great respect to you again, thank you for your advice, best wishes to you.

Re: Spectrum01488-24R1 (Exploring the cecal microbial community associated with fat deposition in sheep and its possible pathways of action)

Dear Dr. Weimin Wang:

Thank you for the privilege of reviewing your work. Below you will find my comments, instructions from the Spectrum editorial office, and the reviewer comments.

Revision Guidelines

Sincerely,
Jianmin Chai
Editor
Microbiology Spectrum

Reviewer #2 (Comments for the Author):

Overall Review

he revised manuscript demonstrates improvements over the original; however, there remain several areas that would benefit from further refinement. While the study's research framework is clear and the findings hold scientific value, aspects such as transparency in experimental details, completeness in methodological descriptions, and depth in data analysis could be

enhanced. Additionally, the comparison with existing research baselines is somewhat limited, which may understate the study's novelty and practical impact. It is hoped that the authors can address these points to strengthen the scientific rigor and persuasive quality of the paper.

Paper Weaknesses

1. Implementation Details: Limited explanation of certain experimental protocols, especially regarding sequencing parameters and preprocessing, may affect reproducibility.
2. Ablation Studies: The study could benefit from an ablation analysis to isolate the effects of individual microbiota on fat traits, improving the robustness of the reported associations.
3. Baseline Comparisons: A more detailed comparison with previous studies on rumen microbiota would help position the findings within the broader context of microbiota research in livestock.

Paper Weaknesses

1. Insufficient Implementation Details: The manuscript lacks comprehensive details for reproducing the study, particularly in the data processing and statistical pipeline. Clarifying the version of software used and parameter settings would greatly enhance reproducibility.
2. Limited Evaluation and Ablation Studies: There are no ablation studies or evaluations showing the individual contributions of key microbiota on fat deposition. This limits the understanding of how each identified biomarker specifically affects fat deposition traits.
3. Lack of Comparison with Baseline Studies: The manuscript includes minimal comparison with widely recognized baseline studies on microbiota and fat deposition in sheep or other ruminants. This would help situate the findings within the existing literature and provide better context for readers.
4. Statistical Validation and Robustness: Some correlations and model predictions are presented without sufficient validation. More detailed cross-validation results or robustness checks would strengthen confidence in the findings, especially regarding the correlations between specific genera and fat deposition traits.
5. Clarity in Exposition: The manuscript lacks clarity in some sections, particularly in describing complex statistical models like the random forest regression. Including more explanation or step-by-step breakdowns would help readers understand the methodology better.

6. Pathway Analysis Interpretation: Although the study explores possible metabolic pathways (e.g., VFAs-TG pathway), the pathway analysis could benefit from further interpretation or discussion, particularly in relation to supporting literature on metabolic pathways affecting fat deposition in similar animal models.

7. Grammar Mistakes: Here are grammar and spelling corrections for the manuscript. I've included line numbers to make it easy to find the errors.

Line Number Original Text Corrected Text

61 Furthermore, reducing fat increases effectively accelerates muscle growth Furthermore, reducing fat accumulation effectively accelerates muscle growth

81 VFAs regulates fat production VFAs regulate fat production

85 Currently, grain pellet feed is used Currently, grain-pellet feed is used

122 20 sheep were selected at the high and low extremes Twenty sheep were selected at the high and low extremes

182 principal coordinate analysis (PCoA) of microorganisms principal coordinate analysis (PCoA) of microbial communities

187 When $P < 0.05$, the genera were considered to When $P < 0.05$, genera were considered to

198 Table S2 summarized the descriptive statistics Table S2 summarizes the descriptive statistics

394 This study comply with the regulations This study complies with the regulations

Questions to Authors and Suggestions for Rebuttal

1. Could you provide additional details on the versions of software and parameter settings used in your sequencing and statistical analysis pipelines to enhance reproducibility?
2. Could an ablation study or similar analysis be performed to demonstrate the impact of individual microbial biomarkers on fat deposition? This would clarify the specific contributions of each identified biomarker.
3. Are there baseline studies or similar research on the role of microbiota in fat deposition in other animals that the authors could use for comparison? How do your findings align with or differ from these?
4. Could additional cross-validation results or statistical checks be presented to strengthen the robustness of the identified correlations, especially regarding Lachnospiraceae_NK3A20_group and fat deposition?
5. For the VFAs-TG metabolic pathway analysis, could the authors elaborate on any supporting literature that might confirm or clarify these findings?

6. Were there any identified genera or VFAs with potentially indirect influences on fat deposition traits that were not detailed in the main text? If so, additional discussion would be helpful.

Overall Score: 6/10

Justification: The study presents novel findings on the association between cecal microbiota and fat deposition in sheep. However, the manuscript's limitations in reproducibility details, robustness checks, and interpretive clarity need addressing. Improved methodological clarity and additional comparative analysis would enhance the impact of this work.

Overall Review

The revised manuscript demonstrates improvements over the original; however, there remain several areas that would benefit from further refinement. While the study's research framework is clear and the findings hold scientific value, aspects such as transparency in experimental details, completeness in methodological descriptions, and depth in data analysis could be enhanced. Additionally, the comparison with existing research baselines is somewhat limited, which may understate the study's novelty and practical impact. It is hoped that the authors can address these points to strengthen the scientific rigor and persuasive quality of the paper.

Paper Weaknesses

1. **Implementation Details:** Limited explanation of certain experimental protocols, especially regarding sequencing parameters and preprocessing, may affect reproducibility.
2. **Ablation Studies:** The study could benefit from an ablation analysis to isolate the effects of individual microbiota on fat traits, improving the robustness of the reported associations.
3. **Baseline Comparisons:** A more detailed comparison with previous studies on rumen microbiota would help position the findings within the broader context of microbiota research in livestock.

Paper Weaknesses

1. **Insufficient Implementation Details:** The manuscript lacks comprehensive details for reproducing the study, particularly in the data processing and statistical pipeline. Clarifying the version of software used and parameter settings would greatly enhance reproducibility.

2. **Limited Evaluation and Ablation Studies:** There are no ablation studies or evaluations showing the individual contributions of key microbiota on fat deposition. This limits the understanding of how each identified biomarker specifically affects fat deposition traits.

3. **Lack of Comparison with Baseline Studies:** The manuscript includes minimal comparison with widely recognized baseline studies on microbiota and fat deposition

in sheep or other ruminants. This would help situate the findings within the existing literature and provide better context for readers.

4. Statistical Validation and Robustness: Some correlations and model predictions are presented without sufficient validation. More detailed cross-validation results or robustness checks would strengthen confidence in the findings, especially regarding the correlations between specific genera and fat deposition traits.

5. Clarity in Exposition: The manuscript lacks clarity in some sections, particularly in describing complex statistical models like the random forest regression. Including more explanation or step-by-step breakdowns would help readers understand the methodology better.

6. Pathway Analysis Interpretation: Although the study explores possible metabolic pathways (e.g., VFAs-TG pathway), the pathway analysis could benefit from further interpretation or discussion, particularly in relation to supporting literature on metabolic pathways affecting fat deposition in similar animal models.

7. Grammar Mistakes: Here are grammar and spelling corrections for the manuscript. I've included line numbers to make it easy to find the errors.

Line Number	Original Text	Corrected Text
61	Furthermore, reducing fat increases effectively accelerates muscle growth	Furthermore, reducing fat accumulation effectively accelerates muscle growth
81	VFAs regulates fat production	VFAs regulate fat production
85	Currently, grain pellet feed is used	Currently, grain-pellet feed is used
122	20 sheep were selected at the high and low extremes	Twenty sheep were selected at the high and low extremes
182	principal coordinate analysis (PCoA) of microorganisms	principal coordinate analysis (PCoA) of microbial communities
187	When $P < 0.05$, the genera were considered to	When $P < 0.05$, genera were considered to
198	Table S2 summarized the descriptive statistics	Table S2 summarizes the descriptive statistics
394	This study comply with the	This study complies with the

Line Number	Original Text	Corrected Text
	regulations	regulations

Questions to Authors and Suggestions for Rebuttal

1. Could you provide additional details on the versions of software and parameter settings used in your sequencing and statistical analysis pipelines to enhance reproducibility?
2. Could an ablation study or similar analysis be performed to demonstrate the impact of individual microbial biomarkers on fat deposition? This would clarify the specific contributions of each identified biomarker.
3. Are there baseline studies or similar research on the role of microbiota in fat deposition in other animals that the authors could use for comparison? How do your findings align with or differ from these?
4. Could additional cross-validation results or statistical checks be presented to strengthen the robustness of the identified correlations, especially regarding *Lachnospiraceae_NK3A20_group* and fat deposition?
5. For the VFAs-TG metabolic pathway analysis, could the authors elaborate on any supporting literature that might confirm or clarify these findings?
6. Were there any identified genera or VFAs with potentially indirect influences on fat deposition traits that were not detailed in the main text? If so, additional discussion would be helpful.

Overall Score: 6/10

Justification: The study presents novel findings on the association between cecal

microbiota and fat deposition in sheep. However, the manuscript's limitations in reproducibility details, robustness checks, and interpretive clarity need addressing. Improved methodological clarity and additional comparative analysis would enhance the impact of this work.

Response to Reviewer Comments

Overall Review

The revised manuscript demonstrates improvements over the original; however, there remain several areas that would benefit from further refinement. While the study's research framework is clear and the findings hold scientific value, aspects such as transparency in experimental details, completeness in methodological descriptions, and depth in data analysis could be enhanced. Additionally, the comparison with existing research baselines is somewhat limited, which may understate the study's novelty and practical impact. It is hoped that the authors can address these points to strengthen the scientific rigor and persuasive quality of the paper.

Response 1: We gratefully appreciate for your valuable suggestion. We will revise the article based on your recommendations.

Paper Weaknesses

1. Implementation Details: Limited explanation of certain experimental protocols, especially regarding sequencing parameters and preprocessing, may affect reproducibility.

Response 2: We gratefully appreciate for your valuable suggestion. The Materials and Methods section has been supplemented and revised. The specific modifications are as follows: 'After obtaining the original sequencing data, it was assigned according to the unique barcode of the sample, and the barcode and primer sequences were excised. The paired-end sequences were merged using FLASH (version 2.7)(20), and FASTP (version 0.23.4) was used to quality control the Raw Tags to obtain Clean Tags. The UCHIME algorithm was used to detect chimeric sequences, and after removing the chimeric sequences, Effective Tags were obtained for subsequent analysis(21). QIIME2 (version 2023.5.1) was used for subsequent analysis(22). The effective tags were imported using the tools import function in QIIME2, and denoising of valid reads was performed using the DADA2 plugin to generate amplicon sequence variants (ASVs) and their representative sequences with the supplementary parameters "denoise-single --p-trunc-len 400 --p-trim-left 20". A phylogenetic tree was constructed using the phylogeny align-to-tree-mafft-fasttree function, and sample microbial diversity indices were calculated using the diversity core-metrics-phylogenetic function. Species annotation (phylum, class, order, family, and genus) was performed based on the SILVA 138 database (<https://www.arb-silva.de/documentation/release-138/>) using the feature-classifier classify-consensus-blast function with the supplementary parameter "--p-maxaccepts 1". PICRUSt2 (Version 2.3.0) was used to predict the functional pathways of microorganisms based on ASVs characteristic sequences and abundance tables, mainly using Kyoto encyclopedia of genes and genomes (KEGG) and metabolic pathways from all domains of life (MetaCyc) databases(23).

R (Version 4.2.2) was utilized for downstream data processing. The Rarefy() function from the GUniFrac package (Version 1.8) was used to normalize all samples to a standard of 25,000 to remove the influence of differences in sequencing depth. Spearman correlation was used to calculate the

correlation and significance between fat deposition traits and other economic traits of sheep and the pheatmap package (Version 1.0.12) was used to draw the correlation heat map. The vegan package (Version 2.6-4) was used to perform principal coordinate analysis (PCoA) of microorganisms based on Bray distance(24), and ANOSIM, ADONIS, and MRPP were used to test the significance of differences between groups. We constructed a random forest regression model using the randomForest package (Version 4.7-1.1), with the relative abundances of genera in the cecum greater than 0.1% as feature variables and the body mass index (BMI) trait of sheep as the target variable. The randomForest() function was employed for model construction to identify biomarkers associated with fat deposition in sheep(25), and the replicate() function with the supplementary parameter "cv.fold = 10 step = 1.5" was utilized for cross-validation. We utilized the stat_cor() function based on the Spearman method to construct a linear fitting model to assess the correlation between genera and BMI. When $P < 0.05$, the genera were considered to be significantly correlated with BMI, and when $P < 0.1$, the genera were considered to have a significant correlation trend with BMI. These genera were defined as key biomarkers. Used the simper() function from the vegan package (Version 2.6-4) to perform SIMPER analysis on biomarkers. A correlation network graph was constructed based on spearman correlation and visualized using Cytoscape (version 3.10.0). All statistical tests in this study were conducted using the wilcoxon rank sum test, with a significance threshold set at $P < 0.05$.

2. Ablation Studies: The study could benefit from an ablation analysis to isolate the effects of individual microbiota on fat traits, improving the robustness of the reported associations.

Response 3: We gratefully appreciate for your valuable suggestion. Thank you for your suggestion. We recognize that ablation studies could provide deeper insights; however, due to the complexity of gut microbiota and the limitations posed by various influencing factors, we are unable to implement ablation experiments in the current study. As an alternative, we will employ SIMPER analysis to better understand the contribution of biomarker microorganisms to fat traits. In future research, we plan to use metagenomic techniques to identify key microorganisms at the species level and conduct single-strain cultures to further explore the impact of microbiota on fat deposition.

"SIMPER analysis was conducted to assess the contribution of cecal biomarkers to intergroup differences (Supplementary Fig S2). The results indicated that both *Bacteroides* and *UCG_005* had contributions exceeding 0.15, suggesting they may be the primary genera responsible for the observed differences. Additionally, *Romboutsia*, *Turicibacter*, and *Lachnospiraceae_NK3A20_group* also demonstrated relatively high contributions."

3. Baseline Comparisons: A more detailed comparison with previous studies on rumen microbiota would help position the findings within the broader context of microbiota research in livestock.

Response 4: We gratefully appreciate for your valuable suggestion. In this study, we primarily conducted an in-depth investigation of the cecal microbiota in sheep. Therefore, we first described the differences in composition between the rumen and cecum in the discussion. Next, we added a baseline study from our preliminary research in the discussion section, as well as a baseline study on microbiota transplantation in mice, focusing on the identified key biomarkers.

"Moreover, the abundance of *Turicibacter* also showed a gradual increase at the nine developmental time points of sheep from 0 to 180 days(59).

Our preliminary baseline study results indicated that when using sheep feces for microbiota transplantation in mice, the abundance of *Bacteroides* significantly increased. Furthermore, recipient mice that received high-fat sheep feces had higher body weight and relative abundance of *Bacteroides* compared to those that received low-fat sheep feces, which is consistent with our findings in sheep(63).

59. Xu D, Cheng J, Zhang D, Huang K, Zhang Y, Li X, Zhao Y, Zhao L, Wang J, Lin C, Yang X, Zhai R, Cui P, Zeng X, Huang Y, Ma Z, Liu J, Han K, Liu X, Yang F, Tian H, Weng X, Zhang X, Wang W. 2023. Relationship between hindgut microbes and feed conversion ratio in Hu sheep and microbial longitudinal development. *J Anim Sci* 101.

63. Cheng J, Zhang X, Zhang D, Zhang Y, Li X, Zhao Y, Xu D, Zhao L, Li W, Wang J, Zhou B, Lin C, Yang X, Zhai R, Cui P, Zeng X, Huang Y, Ma Z, Liu J, Wang W. 2022. Sheep fecal transplantation affects growth performance in mouse models by altering gut microbiota. *J Anim Sci* 100."

Paper Weaknesses

1. Insufficient Implementation Details: The manuscript lacks comprehensive details for reproducing the study, particularly in the data processing and statistical pipeline. Clarifying the version of software used and parameter settings would greatly enhance reproducibility.

Response 5: We gratefully appreciate for your valuable suggestion. The analysis pipeline, software versions, and parameter settings for the data processing section have been supplemented.

"After obtaining the original sequencing data, it was assigned according to the unique barcode of the sample, and the barcode and primer sequences were excised. The paired-end sequences were merged using FLASH (version 2.7)(20), and FASTP (version 0.23.4) was used to quality control the Raw Tags to obtain Clean Tags. The UCHIME algorithm was used to detect chimeric sequences, and after removing the chimeric sequences, Effective Tags were obtained for subsequent analysis(21). QIIME2 (version 2023.5.1) was used for subsequent analysis(22). The effective tags were imported using the tools import function in QIIME2, and denoising of valid reads was performed using the DADA2 plugin to generate amplicon sequence variants (ASVs) and their representative sequences with the supplementary parameters "denoise-single --p-trunc-len 400 --p-trim-left 20". A phylogenetic tree was constructed using the phylogeny align-to-tree-mafft-fasttree function, and sample microbial diversity indices were calculated using the diversity core-metrics-phylogenetic function. Species annotation (phylum, class, order, family, and genus) was performed based on the SILVA 138 database (<https://www.arb-silva.de/documentation/release-138/>) using the feature-classifier classify-consensus-blast function with the supplementary parameter "--p-maxaccepts 1". PICRUST2 (Version 2.3.0) was used to predict the functional pathways of microorganisms based on ASVs characteristic sequences and abundance tables, mainly using Kyoto encyclopedia of genes and genomes (KEGG) and metabolic pathways from all domains of life (MetaCyc) databases(23).

R (Version 4.2.2) was utilized for downstream data processing. The Rarefy() function from the GUniFrac package (Version 1.8) was used to normalize all samples to a standard of 25,000 to remove the influence of differences in sequencing depth. Spearman correlation was used to calculate the correlation and significance between fat deposition traits and other economic traits of sheep and the pheatmap package (Version 1.0.12) was used to draw the correlation heat map. The vegan package (Version 2.6-4) was used to perform principal coordinate analysis (PCoA) of microorganisms based on Bray distance(24), and ANOSIM, ADONIS, and MRPP were used to test the significance of

differences between groups. We constructed a random forest regression model using the randomForest package (Version 4.7-1.1), with the relative abundances of genera in the cecum greater than 0.1% as feature variables and the body mass index (BMI) trait of sheep as the target variable. The randomForest() function was employed for model construction to identify biomarkers associated with fat deposition in sheep(25), and the replicate() function with the supplementary parameter "cv.fold = 10 step = 1.5" was utilized for cross-validation. We utilized the stat_cor() function based on the Spearman method to construct a linear fitting model to assess the correlation between genera and BMI. When $P < 0.05$, the genera were considered to be significantly correlated with BMI, and when $P < 0.1$, the genera were considered to have a significant correlation trend with BMI. These genera were defined as key biomarkers. Used the simper() function from the vegan package (Version 2.6-4) to perform SIMPER analysis on biomarkers. A correlation network graph was constructed based on spearman correlation and visualized using Cytoscape (version 3.10.0). All statistical tests in this study were conducted using the wilcoxon rank sum test, with a significance threshold set at $P < 0.05$."

2. Limited Evaluation and Ablation Studies: There are no ablation studies or evaluations showing the individual contributions of key microbiota on fat deposition. This limits the understanding of how each identified biomarker specifically affects fat deposition traits.

Response 6: We gratefully appreciate for your valuable suggestion. Your suggestion is very valid. However, due to the complexity of gut microbiota and the limitations of 16S technology, we currently lack ablation experiments in this study. As an alternative, we used the SIMPER analysis method to calculate the contributions of different biomarkers. In future research, we plan to implement more detailed and in-depth experimental designs.

"SIMPER analysis was conducted to assess the contribution of cecal biomarkers to intergroup differences (Supplementary Fig S2). The results indicated that both *Bacteroides* and *UCG_005* had contributions exceeding 0.15, suggesting they may be the primary genera responsible for the observed differences. Additionally, *Romboutsia*, *Turicibacter*, and *Lachnospiraceae_NK3A20_group* also demonstrated relatively high contributions."

3. Lack of Comparison with Baseline Studies: The manuscript includes minimal comparison with widely recognized baseline studies on microbiota and fat deposition in sheep or other ruminants. This would help situate the findings within the existing literature and provide better context for readers.

Response 7: We gratefully appreciate for your valuable suggestion. We added a baseline study of nine growth and development time points in sheep to the discussion section, along with a baseline study on gut microbiota transplantation and fat development.

"Moreover, the abundance of *Turicibacter* also showed a gradual increase at the nine developmental time points of sheep from 0 to 180 days(59).

Our preliminary baseline study results indicated that when using sheep feces for microbiota transplantation in mice, the abundance of *Bacteroides* significantly increased. Furthermore, recipient mice that received high-fat sheep feces had higher body weight and relative abundance of *Bacteroides* compared to those that received low-fat sheep feces, which is consistent with our findings in sheep(63).

59. Xu D, Cheng J, Zhang D, Huang K, Zhang Y, Li X, Zhao Y, Zhao L, Wang J, Lin C, Yang X, Zhai R, Cui P, Zeng X, Huang Y, Ma Z, Liu J, Han K, Liu X, Yang F, Tian H, Weng X, Zhang X,

Wang W. 2023. Relationship between hindgut microbes and feed conversion ratio in Hu sheep and microbial longitudinal development. *J Anim Sci* 101.

63. Cheng J, Zhang X, Zhang D, Zhang Y, Li X, Zhao Y, Xu D, Zhao L, Li W, Wang J, Zhou B, Lin C, Yang X, Zhai R, Cui P, Zeng X, Huang Y, Ma Z, Liu J, Wang W. 2022. Sheep fecal transplantation affects growth performance in mouse models by altering gut microbiota. *J Anim Sci* 100."

4. Statistical Validation and Robustness: Some correlations and model predictions are presented without sufficient validation. More detailed cross-validation results or robustness checks would strengthen confidence in the findings, especially regarding the correlations between specific genera and fat deposition traits.

Response 8: We gratefully appreciate for your valuable suggestion. To address your concerns regarding statistical validation and robustness, we have verified the correlations between biomarkers and fat deposition traits using the Pearson correlation method. The results confirm the reliability and consistency of the observed correlations. We have included a detailed description of the validation process and the corresponding results in the revised manuscript.

"The Pearson method was used to analyze the correlations between 15 biomarkers and fat deposition traits. The results were consistent with the linear fitting outcomes, showing that biomarkers were significantly correlated with fat deposition traits ($P < 0.05$) (Supplementary Fig S3)."

5. Clarity in Exposition: The manuscript lacks clarity in some sections, particularly in describing complex statistical models like the random forest regression. Including more explanation or step-by-step breakdowns would help readers understand the methodology better.

Response 9: We gratefully appreciate for your valuable suggestion. The materials and methods section for the random forest analysis has been supplemented.

"We constructed a random forest regression model using the randomForest package (Version 4.7-1.1), with the relative abundances of genera in the cecum greater than 0.1% as feature variables and the body mass index (BMI) trait of sheep as the target variable. The randomForest() function was employed for model construction to identify biomarkers associated with fat deposition in sheep(25), and the replicate() function with the supplementary parameter "cv.fold = 10 step = 1.5" was utilized for cross-validation."

6. Pathway Analysis Interpretation: Although the study explores possible metabolic pathways (e.g., VFAs-TG pathway), the pathway analysis could benefit from further interpretation or discussion, particularly in relation to supporting literature on metabolic pathways affecting fat deposition in similar animal models.

Response 10: We gratefully appreciate for your valuable suggestion. A discussion of the supporting literature on metabolic pathways by which microbes influence host fat deposition has been added to the Discussion section.

"Research results also showed that microorganisms might have regulated the host's metabolism through the gut-brain axis, with the gut microbiota influencing the bidirectional communication between the gut and the brain(53). Previous research results showed that intestinal flora and their metabolites could lead

to obesity(54). It was also related to the strong energy acquisition ability of gut microorganisms associated with obesity(55). At the same time, there might have been a process of bacterial cross-feeding between positively correlated biomarkers, where one microbial community provided nutrients for another(56). The research results on mice showed that some gut microorganisms were significantly negatively correlated with body weight and serum TG, which was similar to our results(57).

53. Rhee SH, Pothoulakis C, Mayer EA. 2009. Principles and clinical implications of the brain-gut-enteric microbiota axis. *Nat Rev Gastroenterol Hepatol* 6:306-14.

54. Gralka E, Luchinat C, Tenori L, Ernst B, Thurnheer M, Schultes B. 2015. Metabolomic fingerprint of severe obesity is dynamically affected by bariatric surgery in a procedure-dependent manner. *Am J Clin Nutr* 102:1313-22.

55. Turnbaugh PJ, Ley RE, Mahowald MA, Magrini V, Mardis ER, Gordon JI. 2006. An obesity-associated gut microbiome with increased capacity for energy harvest. *Nature* 444:1027-31.

56. Ríos-Covián D, Ruas-Madiedo P, Margolles A, Gueimonde M, de Los Reyes-Gavilán CG, Salazar N. 2016. Intestinal Short Chain Fatty Acids and their Link with Diet and Human Health. *Front Microbiol* 7:185.

57. Feng J, Ma H, Huang Y, Li J, Li W. 2022. Ruminococcaceae_UCG-013 Promotes Obesity Resistance in Mice. *Biomedicines* 10.".

7. Grammar Mistakes:Here are grammar and spelling corrections for the manuscript. I've included line numbers to make it easy to find the errors.

Response 11: We gratefully appreciate for your valuable suggestion. The incorrect area has been modified.

Questions to Authors and Suggestions for Rebuttal

1. Could you provide additional details on the versions of software and parameter settings used in your sequencing and statistical analysis pipelines to enhance reproducibility?

Response 12: We gratefully appreciate for your valuable suggestion. Detailed supplements to the analysis pipeline have been provided in the Materials and Methods.

"After obtaining the original sequencing data, it was assigned according to the unique barcode of the sample, and the barcode and primer sequences were excised. The paired-end sequences were merged using FLASH (version 2.7)(20), and FASTP (version 0.23.4) was used to quality control the Raw Tags to obtain Clean Tags. The UCHIME algorithm was used to detect chimeric sequences, and after removing the chimeric sequences, Effective Tags were obtained for subsequent analysis(21). QIIME2 (version 2023.5.1) was used for subsequent analysis(22). The effective tags were imported using the tools import function in QIIME2, and denoising of valid reads was performed using the DADA2 plugin to generate amplicon sequence variants (ASVs) and their representative sequences with the supplementary parameters "denoise-single --p-trunc-len 400 --p-trim-left 20". A phylogenetic tree was constructed using the phylogeny align-to-tree-mafft-fasttree function, and sample microbial diversity indices were calculated using the diversity core-metrics-phylogenetic function. Species annotation (phylum, class, order, family, and genus) was performed based on the SILVA 138 database

(<https://www.arb-silva.de/documentation/release-138/>) using the feature-classifier classify-consensus-blast function with the supplementary parameter "--p-maxaccepts 1". PICRUSt2 (Version 2.3.0) was used to predict the functional pathways of microorganisms based on ASVs characteristic sequences and abundance tables, mainly using Kyoto encyclopedia of genes and genomes (KEGG) and metabolic pathways from all domains of life (MetaCyc) databases(23).

R (Version 4.2.2) was utilized for downstream data processing. The Rarefy() function from the GUniFrac package (Version 1.8) was used to normalize all samples to a standard of 25,000 to remove the influence of differences in sequencing depth. Spearman correlation was used to calculate the correlation and significance between fat deposition traits and other economic traits of sheep and the pheatmap package (Version 1.0.12) was used to draw the correlation heat map. The vegan package (Version 2.6-4) was used to perform principal coordinate analysis (PCoA) of microorganisms based on Bray distance(24), and ANOSIM, ADONIS, and MRPP were used to test the significance of differences between groups. We constructed a random forest regression model using the randomForest package (Version 4.7-1.1), with the relative abundances of genera in the cecum greater than 0.1% as feature variables and the body mass index (BMI) trait of sheep as the target variable. The randomForest() function was employed for model construction to identify biomarkers associated with fat deposition in sheep(25), and the replicate() function with the supplementary parameter "cv.fold = 10 step = 1.5" was utilized for cross-validation. We utilized the stat_cor() function based on the Spearman method to construct a linear fitting model to assess the correlation between genera and BMI. When $P < 0.05$, the genera were considered to be significantly correlated with BMI, and when $P < 0.1$, the genera were considered to have a significant correlation trend with BMI. These genera were defined as key biomarkers. Used the simper() function from the vegan package (Version 2.6-4) to perform SIMPER analysis on biomarkers. A correlation network graph was constructed based on spearman correlation and visualized using Cytoscape (version 3.10.0). All statistical tests in this study were conducted using the wilcoxon rank sum test, with a significance threshold set at $P < 0.05$."

2. Could an ablation study or similar analysis be performed to demonstrate the impact of individual microbial biomarkers on fat deposition? This would clarify the specific contributions of each identified biomarker.

Response 13: We gratefully appreciate for your valuable suggestion. Thank you for your suggestion. We recognize that ablation studies could provide deeper insights; however, due to the complexity of gut microbiota and the limitations posed by various influencing factors, we are unable to implement ablation experiments in the current study. As an alternative, we will employ SIMPER analysis to better understand the contribution of biomarker microorganisms to fat traits. In future research, we plan to use metagenomic techniques to identify key microorganisms at the species level and conduct single-strain cultures to further explore the impact of microbiota on fat deposition.

"SIMPER analysis was conducted to assess the contribution of cecal biomarkers to intergroup differences (Supplementary Fig S2). The results indicated that both *Bacteroides* and *UCG_005* had contributions exceeding 0.15, suggesting they may be the primary genera responsible for the observed differences. Additionally, *Romboutsia*, *Turicibacter*, and *Lachnospiraceae_NK3A20_group* also demonstrated relatively high contributions."

3. Are there baseline studies or similar research on the role of microbiota in fat deposition in other animals that the authors could use for comparison? How do your findings align with or differ from these?

Response 14: We gratefully appreciate for your valuable suggestion. In this study, we primarily conducted an in-depth investigation of the cecal microbiota in sheep. Therefore, we first described the differences in composition between the rumen and cecum in the discussion. Next, we added a baseline study from our preliminary research in the discussion section, as well as a baseline study on microbiota transplantation in mice, focusing on the identified key biomarkers. The above baseline study is consistent with the results of this study.

"Moreover, the abundance of *Turicibacter* also showed a gradual increase at the nine developmental time points of sheep from 0 to 180 days(59).

Our preliminary baseline study results indicated that when using sheep feces for microbiota transplantation in mice, the abundance of *Bacteroides* significantly increased. Furthermore, recipient mice that received high-fat sheep feces had higher body weight and relative abundance of *Bacteroides* compared to those that received low-fat sheep feces, which is consistent with our findings in sheep(63).

59. Xu D, Cheng J, Zhang D, Huang K, Zhang Y, Li X, Zhao Y, Zhao L, Wang J, Lin C, Yang X, Zhai R, Cui P, Zeng X, Huang Y, Ma Z, Liu J, Han K, Liu X, Yang F, Tian H, Weng X, Zhang X, Wang W. 2023. Relationship between hindgut microbes and feed conversion ratio in Hu sheep and microbial longitudinal development. *J Anim Sci* 101.

63. Cheng J, Zhang X, Zhang D, Zhang Y, Li X, Zhao Y, Xu D, Zhao L, Li W, Wang J, Zhou B, Lin C, Yang X, Zhai R, Cui P, Zeng X, Huang Y, Ma Z, Liu J, Wang W. 2022. Sheep fecal transplantation affects growth performance in mouse models by altering gut microbiota. *J Anim Sci* 100."

4. Could additional cross-validation results or statistical checks be presented to strengthen the robustness of the identified correlations, especially regarding *Lachnospiraceae_NK3A20_group* and fat deposition?

Response 15: We gratefully appreciate for your valuable suggestion. To address your concerns regarding statistical validation and robustness, we have verified the correlations between biomarkers and fat deposition traits using the Pearson correlation method. The results confirm the reliability and consistency of the observed correlations. We have included a detailed description of the validation process and the corresponding results in the revised manuscript.

"The Pearson method was used to analyze the correlations between 15 biomarkers and fat deposition traits. The results were consistent with the linear fitting outcomes, showing that biomarkers were significantly correlated with fat deposition traits ($P < 0.05$) (Supplementary Fig S3)."

5. For the VFAs-TG metabolic pathway analysis, could the authors elaborate on any supporting literature that might confirm or clarify these findings?

Response 16: We gratefully appreciate for your valuable suggestion. The discussion section has been supplemented and relevant literature has been added.

"54. Gralka E, Luchinat C, Tenori L, Ernst B, Thurnheer M, Schultes B. 2015. Metabolomic fingerprint of severe obesity is dynamically affected by bariatric surgery in a procedure-dependent manner. *Am J Clin Nutr* 102:1313-22.

56. Ríos-Covián D, Ruas-Madiedo P, Margolles A, Gueimonde M, de Los Reyes-Gavilán CG, Salazar N. 2016. Intestinal Short Chain Fatty Acids and their Link with Diet and Human Health. *Front Microbiol* 7:185.

57. Feng J, Ma H, Huang Y, Li J, Li W. 2022. Ruminococcaceae_UCG-013 Promotes Obesity Resistance in Mice. *Biomedicines* 10."

6. Were there any identified genera or VFAs with potentially indirect influences on fat deposition traits that were not detailed in the main text? If so, additional discussion would be helpful.

Response 17: We gratefully appreciate for your valuable suggestion. This study currently identified 15 key biomarkers, among which *Lachnospiraceae_NK3A20_group*, *Turicibacter*, and *Bacteroides* may play a synergistic role and are negatively correlated with fat deposition in sheep. This result has been supplemented in detail in the Results and Discussion sections.

Justification: The study presents novel findings on the association between cecal microbiota and fat deposition in sheep. However, the manuscript's limitations in reproducibility details, robustness checks, and interpretive clarity need addressing. Improved methodological clarity and additional comparative analysis would enhance the impact of this work.

Response 18: Thank you very much for your guidance and suggestions on the manuscript. The article has been revised and improved, and I would like to express my gratitude once again.

Great respect to you again, thank you for your advice, best wishes to you.

Re: Spectrum01488-24R2 (Exploring the cecal microbial community associated with fat deposition in sheep and its possible pathways of action)

Dear Dr. Weimin Wang:

Thank you for the privilege of reviewing your work. Below you will find my comments, instructions from the Spectrum editorial office, and the reviewer comments.

Revision Guidelines

Sincerely,
Jianmin Chai
Editor
Microbiology Spectrum

Reviewer #2 (Public repository details (Required)):

The study does include large sequencing datasets (over 4 million raw tags), but the document lacks details about public deposition. For confirmation, readers would need to check supplementary materials or contact the authors directly.

Reviewer #2 (Comments for the Author):

Comments and Suggestions for the Authors:

Clarity and Repetition in Text:

Multiple sections of the abstract and results show repetition (e.g., "Spearman correlation analysis..." and "IMPORTANCE" sections duplicated). Streamline the text to avoid redundancy and improve readability. Check formatting inconsistencies (e.g., incomplete figure/table labels like "Fig 3e" missing panel references, and phrases such as "Supplementary S2)" missing table numbers).

Sample Grouping and Statistical Rigor:

Clarify how sheep were stratified into high (HB), medium (MB), and low (LB) fat deposition groups. The study mentions using BMI to sort 303 sheep and selecting 20 each at extremes and median, but justification for the "20% test population" needs elaboration. The non-significant differences between MB and other groups in microbial structure (Fig 3d) may reflect biological continuity or statistical limitations. Discuss potential reasons (e.g., sample size, overlap in phenotypes).
Mechanistic Insights:

While the correlation between microbial biomarkers (e.g., Lachnospiraceae_NK3A20_group) and acetic acid/TG levels is noted, deeper exploration of how these microbes influence fat deposition (e.g., enzymatic pathways, VFA production) would strengthen conclusions. In the introduction, the hypothesis states "cecal microbiota influences fat deposition via VFAs-TG pathways," but the results primarily show correlations. Validate causality through additional analyses (e.g., mediation analysis or functional experiments) or explicitly frame findings as associative.

Data Presentation:

Provide visual clarity for key results:

Include PCoA plots (Fig 3d) and microbial composition bar charts (Fig 3e) in the main text or ensure supplementary figures are referenced correctly. Report effect sizes alongside P-values (e.g., Firmicutes/Bacteroidota abundance differences between LB and HB groups).

Clarify methods for BMI calculation: Body length is used in the formula, but "m²" may not be standard for sheep. Justify this metric or cite prior validation.

Technical Validation:

Specify quality control steps for 16S sequencing (e.g., ASV filtering thresholds, removal of low-depth samples). Mention whether rarefaction was applied before alpha/beta diversity analyses. For the random forest regression model, report cross-validation accuracy (e.g., R²) and ensure biomarkers were not overfitted (e.g., permutation testing).

Broader Implications:

Emphasize how cecal microbial modulation could improve livestock leanness in practice (e.g., dietary interventions, probiotics). Link findings to existing literature on hindgut microbiota in ruminants. Address potential confounding factors (e.g., diet standardization, batch effects in sequencing) that might influence microbiota composition.

Language and Typos:

Minor errors need correction (e.g., "Husion High-Fidelity PCR" → "Husion@", "row tags" → "raw tags"). Improve sentence structure for clarity (e.g., "The feeding regimen and environmental conditions were consistent throughout the study. 20% of the population..." → "From the initial 303 sheep, 60 [20%] were selected...").

Strengths to Highlight:

The study provides novel insights into the neglected role of the cecum in ruminant fat metabolism. Integrates microbial sequencing, VFA profiling, and host biochemistry to propose a cohesive pathway (VFAs-TG). Application of machine learning (random forest) to identify microbial biomarkers adds methodological rigor.

Final Recommendation:

Revise to address mechanistic gaps, statistical clarity, and text redundancies. With these improvements, the study offers valuable contributions to understanding microbial drivers of fat deposition in sheep.

Response to Reviewer Comments

Reviewer #2 (Public repository details (Required)):

The study does include large sequencing datasets (over 4 million raw tags), but the document lacks details about public deposition. For confirmation, readers would need to check supplementary materials or contact the authors directly.

Response 1: We gratefully appreciate for your valuable suggestion. The raw sequencing data generated in this study have been uploaded to the NCBI Sequence Read Archive (submission ID: SUB14387386, BioProject ID: PRJNA1101910). We will release these data prior to publication.

- ✓ BioProject: **Processed**
PRJNA1101910 : Exploring the cecal microbial community associated with fat deposition in sheep and its possible pathways of action
- ✓ BioSample: **Processed**
Successfully loaded
(60 objects)
Download attributes file with BioSample accessions
- ✓ SRA: **Processed**
(60 objects)
Download metadata file with SRA accessions
View and manage my SRA submission data

Reviewer #2 (Comments for the Author):

Comments and Suggestions for the Authors:

Clarity and Repetition in Text:

Multiple sections of the abstract and results show repetition (e.g., "Spearman correlation analysis..." and "IMPORTANCE" sections duplicated). Streamline the text to avoid redundancy and improve readability. Check formatting inconsistencies (e.g., incomplete figure/table labels like "Fig 3e" missing panel references, and phrases such as "Supplementary S2)" missing table numbers).

Response 2: We gratefully appreciate for your valuable suggestion. We have revised the ABSTRACT, IMPORTANCE, and RESULTS, simplified the text, and removed redundant parts.

Figure 3e has been split and referenced separately. The specific modifications are as follows: 'We observe that at the phylum level, Firmicutes abundance decreased and Bacteroidota abundance increased across the LB, MB, and HB groups (Figure 3f). The Firmicutes in the LB group are significantly higher than those in the HB group, while the Bacteroidota are significantly lower than those in the HB group ($P < 0.05$) (Supplementary Table S4). The Firmicutes/Bacteroidota value of the

LB group was significantly higher than that of the HB group ($P < 0.05$) (Supplementary Table S4). The relative abundance of Lachnospiraceae at the family level shows a decreasing trend, and species composition was similar between groups (Figure 3g).

All supplementary files have been checked and the insertion position of Supplementary S2 has been changed to: 'A total of 303 Hu sheep were used in this cross-sectional study. Descriptive statistics of fat deposition traits are summarized in Supplementary Table S2. BMI was used as a comprehensive evaluation index for sheep fat deposition traits. Based on BMI, individuals with high (HB) and low (LB) extreme BMI, as well as control BMI (MB), were selected (n = 60, 20% of the cohort). The average

BMI of the large tested group (303 Hu sheep) was 85.28 kg/m² (Supplementary Table S2), and there were significant differences in BMI among the three groups ($P < 0.01$) (Figure 1a).'

Sample Grouping and Statistical Rigor:

Clarify how sheep were stratified into high (HB), medium (MB), and low (LB) fat deposition groups. The study mentions using BMI to sort 303 sheep and selecting 20 each at extremes and median, but justification for the "20% test population" needs elaboration. The non-significant differences between MB and other groups in microbial structure (Fig 3d) may reflect biological continuity or statistical limitations. Discuss potential reasons (e.g., sample size, overlap in phenotypes).

Response 3: We gratefully appreciate for your valuable suggestion. BMI was used to sort the population and twenty sheep were selected at the high and low extremes and 20 sheep at the median were taken as the control group, supplemented in Materials Methods.

Selecting 20% of the population ensured adequate statistical power while balancing experimental costs and resource availability. The sample size of 60 sheep (20 per group) provided enough data to identify significant biological variations without overextending the financial and logistical requirements of sequencing and biochemical assays. By specifically selecting individuals at the high and low extremes of BMI (along with a middle BMI reference group), we aimed to enhance the ability to discern distinct differences in microbiota composition and metabolic traits. The inclusion of the middle BMI group provided a baseline for comparison between the extremes. We will incorporate the above justification in the revised Materials and Methods section to ensure clarity and address the reviewer's concern.

Thank you for highlighting the non-significant microbial structure differences between the MB group and the other extreme groups (HB and LB) in Figure 3d. We agree that this observation requires further discussion, and we will address this in the revised manuscript. Below are the key factors we believe may contribute to this result: The middle BMI group (MB) represents a biologically intermediate population between individuals from the extreme BMI groups (HB and LB). This suggests that the microbiota composition of the MB group may reflect a transitional state, sharing microbial features with both extremes. Such biological continuity could reduce the statistical separation compared with the more distinct microbiota profiles of the HB and LB groups. Although each group contains an equal number of samples ($n = 20$), it is possible that the BMI distribution within the MB group includes a wider range of phenotypes compared to the more narrowly defined extreme groups. This may result in some overlap in microbiota features between the MB and extreme groups.

Your suggestions have been of great help to us, and we will include a larger group in our future research.

Mechanistic Insights:

While the correlation between microbial biomarkers (e.g., Lachnospiraceae_NK3A20_group) and acetic acid/TG levels is noted, deeper exploration of how these microbes influence fat deposition (e.g., enzymatic pathways, VFA production) would strengthen conclusions. In the introduction, the hypothesis states "cecal microbiota influences fat deposition via VFAs-TG pathways," but the results

primarily show correlations. Validate causality through additional analyses (e.g., mediation analysis or functional experiments) or explicitly frame findings as associative.

Response 4: We gratefully appreciate for your valuable suggestion. On validating causality hypothesis: To address the suggestion of validating causality, we have conducted mediation analyses to establish the role of TG as intermediary factors influencing fat deposition. The results, presented in Supplementary Table S6, demonstrate that TG acts as a significant mediator, supporting the proposed pathway hypothesis. We have clarified these findings in the revised Results sections. The method for the mediation analysis was supplemented in the materials and methods. The specific modifications are as follows: 'Conducted mediation analysis using Model 4 in the PROCESS (Version 4.1) plugin of IBM SPSS Statistics 27, employing the Bootstrap method (5000 iterations) for mediation effect testing.', 'Further mediation analysis, as shown in Supplementary Table S6, demonstrated that the direct effect of the *Lachnospiraceae_NK3A20_group* on BMI was -684.73, with a confidence interval that did not include 0, indicating a significant direct effect. Moreover, the mediating effect value via TG was -114.62, also with a confidence interval excluding 0, suggesting that TG plays a significant mediating role in this mechanism.'. While our current study provides evidence for significant correlations between microbial markers and fat deposition traits. However, we agree with your point that the lack of validation experiments is a limitation of this study. Since we identified the taxa at the genus level through 16S rDNA sequencing, we will use metagenomic techniques to identify the biomarkers at the species level in future research. Additionally, we will conduct experimental validation through single strain cultivation and microbiota transplantation. We have revised the Discussion section to elaborate on the study limitations, as well as the need for further experimental validation. The specific modifications are as follows: 'Overall, this study contributes to advancing our understanding of the regulatory pathways underlying fat deposition traits in sheep, which could accelerate the development of efficient microbial additives. Despite these findings, certain limitations should be acknowledged. While we have identified some potential biomarkers and explored possible mechanisms, further functional validation, such as single-strain isolation and culture or microbiota transplantation, is still required.'.

Supplementary Table S6. Analysis of the mediating effect of *Lachnospiraceae_NK3A20_group* and BMI.

Model effect	Effect	Boot SE	BootLLCI	BootULCI	Effect ratio
Total effect	-799.35	202.42	-1204.55	-394.16	100%
Direct effect	-684.73	198.22	-1081.67	-287.80	85.66%
Indirect effect	-114.62	104.50	-393.80	-0.97	14.34%

Data Presentation:

Provide visual clarity for key results:

Include PCoA plots (Fig 3d) and microbial composition bar charts (Fig 3e) in the main text or ensure supplementary figures are referenced correctly. Report effect sizes alongside P-values (e.g., Firmicutes/Bacteroidota abundance differences between LB and HB groups).

Response 5: We gratefully appreciate for your valuable suggestion. We have rewritten and split the figures to ensure they can be correctly referenced in the text, and we have modified the figure legend to: "Figure 3. Differences in cecal microorganisms of sheep with different fat deposits. (a) VENN plot of ASVs. (b) Spearman correlation of ASVs between groups. (c) Alpha diversity difference analysis, Shannon and Chao1 index. (d) PCoA analysis based on bray distance. The bottom box plot shows the values of PCoA1, while the right box plot shows the values of PCoA2. *, $P < 0.05$; **, $P < 0.01$; ***, $P < 0.001$. (e) Community structure difference testing (ANOSIM, ADONIS, and MRPP). (f) Relative abundance of phylum-level and (g) family-level in different groupings. (h) Biomarkers identified based on RF regression model and tenfold cross-validation. (i) Linear fitting correlation analysis between biomarkers and BMI."

The differences described in the text have been supplemented with P-values, and all difference results are included in Supplementary Table S4.

Supplementary Table S4. Analysis of differences in phylum and family among different groups.

Phylum	LB	MB	HB
Firmicutes	0.6746±0.0719 ^a	0.6416±0.0675 ^{ab}	0.6127±0.0802 ^b
Bacteroidota	0.2003±0.0482 ^b	0.2361±0.048 ^{ab}	0.2746±0.0701 ^a
Spirochaetota	0.0172±0.0143	0.0321±0.0276	0.0306±0.0289
Verrucomicrobiota	0.0285±0.0375	0.0199±0.016	0.0217±0.0229
Desulfobacterota	0.0215±0.0141	0.0197±0.0103	0.0211±0.015
Firmicutes/Bacteroidota	3.6187±1.21 ^a	2.8989±0.9961 ^{ab}	2.4641±1.1107 ^b
Family	LB	MB	HB
Lachnospiraceae	0.1601±0.0423	0.1507±0.0295	0.1362±0.028
Oscillospiraceae	0.1302±0.0188	0.1216±0.0201	0.1187±0.0233
Eubacterium	0.0667±0.0213	0.0823±0.037	0.076±0.0263
Rikenellaceae	0.0554±0.0226 ^b	0.0789±0.0303 ^a	0.0858±0.0358 ^a
Prevotellaceae	0.0624±0.0447	0.0663±0.0419	0.0739±0.0559
Bacteroidaceae	0.0379±0.0126 ^b	0.05±0.0229 ^{ab}	0.0593±0.0197 ^a
UCG-010	0.0389±0.0094	0.0459±0.0101	0.0455±0.0141
Ruminococcaceae	0.044±0.0092	0.042±0.0067	0.0441±0.0123
Christensenellaceae	0.0467±0.0122 ^a	0.0429±0.0133 ^{ab}	0.0369±0.0094 ^b
Monoglobaceae	0.0405±0.0103	0.0434±0.0104	0.0421±0.0157

Note: Statistical data are expressed as mean ± standard deviation. $P < 0.05$ indicates statistical significance. Different lowercase letters in the same row indicate significant differences between different group ($P < 0.05$).

Clarify methods for BMI calculation: Body length is used in the formula, but "m²" may not be standard for sheep. Justify this metric or cite prior validation.

Response 6: We gratefully appreciate for your valuable suggestion. The Materials and Methods section has been revised to "Based on previous reports, the formula for sheep BMI is weight (kg) divided by the square of the body length (m)(18)."

18. Zhang Y, Zhang X, Li C, Tian H, Weng X, Lin C, Zhang D, Zhao Y, Li X, Cheng J, Zhao L, Xu D, Yang X, Jiang Z, Li F, Wang W. 2024. Rumen microbiome and fat deposition in sheep: insights from a bidirectional mendelian randomization study. *NPJ Biofilms Microbiomes* 10:129.

Technical Validation:

Specify quality control steps for 16S sequencing (e.g., ASV filtering thresholds, removal of low-depth samples). Mention whether rarefaction was applied before alpha/beta diversity analyses. For the random forest regression model, report cross-validation accuracy (e.g., R^2) and ensure biomarkers were not overfitted (e.g., permutation testing).

Response 7: We gratefully appreciate for your valuable suggestion. My filtering criteria for the 16S sequencing data are based on the following parameters: I trimmed the first 20 bases from each sequence using `--p-trim-left 20`, and I truncated each sequence at 400 bases using `--p-trunc-len 400`. Additionally, the DADA2 algorithm was employed to denoise the data and remove sequencing errors, including chimera removal, ensuring the accuracy of the generated feature sequences (ASVs). Supplementary parameters have been added to the Materials and Methods section. The `Rarefy()` function from the GUniFrac package (Version 1.8) was used to normalize all samples to a standard of 25,000 to remove the influence of differences in sequencing depth. Alpha/beta diversity analysis was performed directly using the normalized ASV abundance table. The R^2 of the random forest regression model was calculated and added to Figure 3h. We evaluated the model by performing 500 permutation tests, and the results showed that the model was not overfitting (Supplementary Figure S2), the Results section has been supplemented.

Broader Implications:

Emphasize how cecal microbial modulation could improve livestock leanness in practice (e.g., dietary interventions, probiotics). Link findings to existing literature on hindgut microbiota in ruminants. Address potential confounding factors (e.g., diet standardization, batch effects in sequencing) that might influence microbiota composition.

Response 8: We gratefully appreciate for your valuable suggestion. We have supplemented the discussion section and added references. Furthermore, we implemented several measures during the experiment to minimize potential confounding factors that might influence microbial composition. Throughout the trial, all animals were provided with a completely standardized diet to ensure consistency in dietary components. In addition, DNA extraction and sequencing for all samples were performed in the same batch to avoid potential batch effects. The specific modifications are as follows: The results of this study suggest that modulating the cecal microbiota has significant potential to improve livestock leanness. In practical applications, incorporating high-fiber feed could selectively promote specific microbial taxa associated with lean tissue deposition, thereby enhancing livestock productivity(65). Furthermore, previous studies have shown that probiotic supplementation significantly increases the abundance of Lachnospiraceae(66), which highlights the potential value of Lachnospiraceae_NK3A20_group as a promising target for further research. Furthermore, previous studies have demonstrated that the hindgut microbiota in ruminants plays a critical role in feed fermentation and feed efficiency(60, 67). Our study further reveals a strong association between hindgut microbiota and host fat deposition. This not only confirms the essential role of hindgut

microbiota in host metabolism but also provides a novel perspective for optimizing livestock leanness through the modulation of cecal microbiota.'

60. Xu D, Cheng J, Zhang D, Huang K, Zhang Y, Li X, Zhao Y, Zhao L, Wang J, Lin C, Yang X, Zhai R, Cui P, Zeng X, Huang Y, Ma Z, Liu J, Han K, Liu X, Yang F, Tian H, Weng X, Zhang X, Wang W. 2023. Relationship between hindgut microbes and feed conversion ratio in Hu sheep and microbial longitudinal development. *J Anim Sci* 101.

65. Zhang X, Liu X, Xie K, Pan Y, Liu F, Hou F. 2025. Effects of different fiber levels of energy feeds on rumen fermentation and the microbial community structure of grazing sheep. *BMC Microbiol* 25:180.

66. Feng Y, Gao SJ, Wei RD, Liu T, Fan XP, Han YD, Zhu N. 2021. Effects of probiotics on intestinal flora, inflammation and degree of liver cirrhosis in rats with liver cirrhosis by regulating Wnt/ β -catenin signaling pathway. *J Biol Regul Homeost Agents* 35:25-33.

67. Jiao JZ, W Z, Guan LL, Tan ZL, Han XF, Tang SX, Zhou CS. 2015. Postnatal bacterial succession and functional establishment of hindgut in supplemental feeding and grazing goats. *J Anim Sci* 93:3528-38.

Language and Typos:

Minor errors need correction (e.g., "Husion High-Fidelity PCR" → "Husion®," "row tags" → "raw tags"). Improve sentence structure for clarity (e.g., "The feeding regimen and environmental conditions were consistent throughout the study. 20% of the population..." → "From the initial 303 sheep, 60 [20%] were selected...").

Response 9: We gratefully appreciate for your valuable suggestion. We are very sorry for the error. We have corrected it and changed "20% of the population was used as the test population size (60) for subsequent experiments. BMI was used to sort the population and twenty sheep were selected at the high and low extremes and 20 sheep at the median were taken as the control group." to "From the initial 303 sheep, 60 (20%) were selected as the test population size for subsequent experiments. Based on BMI rankings, 20 sheep from the high and low extremes were selected, while 20 sheep from the median served as the control group."

Strengths to Highlight:

The study provides novel insights into the neglected role of the cecum in ruminant fat metabolism. Integrates microbial sequencing, VFA profiling, and host biochemistry to propose a cohesive pathway (VFAs-TG). Application of machine learning (random forest) to identify microbial biomarkers adds methodological rigor.

Response 10: We gratefully appreciate for your valuable suggestion. Thank you for your recognition and support. Your suggestions have provided tremendous help for our future work. Once again, thank you.

Final Recommendation:

Revise to address mechanistic gaps, statistical clarity, and text redundancies. With these improvements, the study offers valuable contributions to understanding microbial drivers of fat deposition in sheep.

Response 11: Thank you for your valuable feedback and constructive suggestions. We greatly appreciate your comments, which have helped us identify areas to improve our manuscript. In response, we have carefully revised the manuscript to address the mechanistic gaps, enhance statistical clarity, and eliminate redundancies in the text. We believe these revisions have strengthened the overall quality of the study, and we hope it now better highlights the valuable contributions of our work to understanding microbial drivers of fat deposition in sheep. Thank you again for your thoughtful review.

I extend my sincere gratitude to you once again for your valuable advice. Wishing you all the best.

Re: Spectrum01488-24R3 (Exploring the cecal microbial community associated with fat deposition in sheep and its possible pathways of action)

Dear Dr. Weimin Wang:

Your manuscript has been accepted, and I am forwarding it to the ASM production staff for publication. Your paper will first be checked to make sure all elements meet the technical requirements. ASM staff will contact you if anything needs to be revised before copyediting and production can begin. Otherwise, you will be notified when your proofs are ready to be viewed.

Sincerely,
Jianmin Chai
Editor
Microbiology Spectrum

Reviewer #2 (Comments for the Author):

This article has both outstanding strengths and areas for improvement in data analysis:

- ****Strengths****

- ****Diverse and Appropriate Analytical Methods****: The article employs a variety of data analysis methods to comprehensively and deeply explore the relationship between the cecal microbial community and fat deposition in sheep. QIIME2 is used for microbial diversity analysis, which can accurately describe the characteristics of the cecal microbial community in sheep with different fat deposition groups, including differences in community structure, species richness, and diversity. PICRUSt2 is utilized to predict microbial functional pathways, revealing the potential roles of microorganisms in the fat deposition process from a functional perspective. Random forest regression models and linear regression analyses are employed to screen for key microbial biomarkers, effectively identifying microbial taxa closely associated with fat deposition. The comprehensive application of these methods provides a comprehensive and in - depth perspective for the research.

- ****Data Standardization and Comparability****: The raw sequencing data are rigorously processed, including operations such as removing barcode and primer sequences, merging paired - end sequences, quality control, and removing chimeric sequences, ensuring the accuracy and reliability of the data. Meanwhile, the Rarefy() function in R language is used to standardize all

samples to a sequencing depth of 25,000, effectively eliminating the influence of differences in sequencing depth on the analysis results. This makes the different samples more comparable and improves the credibility of the data analysis results.

- **In - Depth Correlation Analysis**: Spearman correlation analysis is used to calculate the correlations between fat deposition traits, other economic traits, and the microbial community, and correlation heatmaps are drawn to visually display the relationships among various factors. In addition, a correlation network diagram based on Spearman correlation is constructed, combined with mediation effect analysis, to deeply explore the complex interactions among cecal microbial biomarkers, cecal metabolites, blood metabolites, and fat deposition traits. This provides strong evidence for revealing the potential mechanisms by which microorganisms affect fat deposition.

- **Weaknesses**

- **Limited Model Explanatory Power**: Although a random forest regression model is constructed to analyze the relationship between cecal microorganisms and fat deposition, the explanatory rate of this model for the variation in sheep fat deposition levels is only 27.05%. This indicates that there may be other important factors not included in the model, or the model construction is not perfect enough. It is necessary to further optimize the model or explore other potential influencing factors to improve the model's explanatory power and predictive ability.

- **Lack of In - Depth Functional Validation Analysis**: Although the functional pathways of microorganisms are predicted through data analysis, there is a lack of further experimental verification of these predicted functions. For example, no in - vitro or in - vivo intervention experiments are conducted on the metabolic functions of key microorganisms, making it impossible to determine the actual functions and mechanisms of microorganisms in the fat deposition process. This affects the reliability of the research results at the functional level to a certain extent.

- **Insufficient Multiple Testing Correction**: A large number of statistical tests, such as difference significance tests and correlation analyses, are carried out in the article, but it does not mention whether multiple testing corrections are performed. When conducting multiple hypothesis tests, if appropriate corrections are not made, the probability of false - positive results will increase, which may lead to misinterpretation of the results. Therefore, it is necessary to consider using appropriate multiple testing correction methods, such as Bonferroni correction or FDR correction, to ensure the accuracy and reliability of the results.